# Inhibitory IL-10-producing CD4+ T cells are T-bet-dependent and facilitate cytomegalovirus persistence via coexpression of arginase-1

Mathew Clement[1,2]*, Kristin Ladell[1], Kelly L Miners[1], Morgan Marsden[1], Lucy Chapman[1], Anna Cardus Figueras[1], Jake Scott[1], Robert Andrews[1,2], Simon Clare[3], Valeriia V Kriukova[4,5,6], Ksenia R Lupyr[4,5,7], Olga V Britanova[5,6,7], David R Withers[8], Simon A Jones[1,2], Dmitriy M Chudakov[4,5,7,9], David A Price[1,2], Ian R Humphreys[1,2]

[1]Division of Infection and Immunity, School of Medicine, Cardiff University, Cardiff, United Kingdom; [2]Systems Immunity Research Institute, School of Medicine, Cardiff University, Cardiff, United Kingdom; [3]Wellcome Sanger Institute, Wellcome Genome Campus, Hinxton, United Kingdom; [4]Center of Life Sciences, Skolkovo Institute of Science and Technology, Moscow, Russian Federation; [5]Genomics of Adaptive Immunity Department, Shemyakin-Ovchinnikov Institute of Bioorganic Chemistry, Russian Academy of Sciences, Moscow, Russian Federation; [6]Institute of Clinical Molecular Biology, Christian-Albrecht-University of Kiel, Kiel, Germany; [7]Institute of Translational Medicine, Center for Precision Genome Editing and Genetic Technologies for Biomedicine, Pirogov Russian National Research Medical University, Moscow, Russian Federation; [8]Institute of Immunology and Immunotherapy, University of Birmingham, Birmingham, United Kingdom; [9]Abu Dhabi Stem Cell Center, Al Muntazah, United Arab Emirates

*For correspondence:
clementm@cardiff.ac.uk

Competing interest: The authors declare that no competing interests exist.

**Abstract** Inhibitory CD4+ T cells have been linked with suboptimal immune responses against cancer and pathogen chronicity. However, the mechanisms that underpin the development of these regulatory cells, especially in the context of ongoing antigen exposure, have remained obscure. To address this knowledge gap, we undertook a comprehensive functional, phenotypic, and transcriptomic analysis of interleukin (IL)-10-producing CD4+ T cells induced by chronic infection with murine cytomegalovirus (MCMV). We identified these cells as clonally expanded and highly differentiated TH1-like cells that developed in a T-bet-dependent manner and coexpressed arginase-1 (Arg1), which promotes the catalytic breakdown of L-arginine. Mice lacking Arg1-expressing CD4+ T cells exhibited more robust antiviral immunity and were better able to control MCMV. Conditional deletion of T-bet in the CD4+ lineage suppressed the development of these inhibitory cells and also enhanced immune control of MCMV. Collectively, these data elucidated the ontogeny of IL-10-producing CD4+ T cells and revealed a previously unappreciated mechanism of immune regulation, whereby viral persistence was facilitated by the site-specific delivery of Arg1.

### Editor's evaluation

This study analyzes the development and functional relevance of IL-10-producing regulatory T cells in a mouse model of cytomegalovirus infection. The results indicate that IL-10-producing CD4+ T cells express genes associated with chronically activated TH1-like cells, undergo clonal expansion,

and inhibit antiviral T cell responses via the secretion of arginase, an enzyme that breaks down an amino acid required for T cell activation and proliferation. These findings reveal a novel and important immunoregulatory mechanism that facilitates viral persistence.

## Introduction

Immune dysregulation occurs during many persistent viral infections. High levels of ongoing viral replication, which characterize human immunodeficiency virus (HIV), hepatitis B virus (HBV), and, in mice, lymphocytic choriomeningitis virus (LCMV), typically lead to T cell exhaustion, defined by impaired effector functions, the expression of inhibitory cytokines and receptors (*Wherry, 2011*), and substantial alterations in cellular gene expression (*Doering et al., 2012*). Moreover, inducible and naturally occurring FoxP3+ regulatory T cells accumulate during many chronic viral infections, presumably to limit excessive immune activation (*Veiga-Parga et al., 2013*), and T helper ($T_H$)1-like cells that express the immunosuppressive cytokine interleukin (IL)-10 can be induced by LCMV (*Parish et al., 2014*), HIV (*Graziosi et al., 1994*), and human/murine cytomegalovirus (HCMV/MCMV) (*Clement et al., 2016*; *Jones et al., 2010*; *Mason et al., 2013*). Evidence from parasitic infections suggests that IL-10-producing $T_H$1-like cells protect against immune pathology, akin to classical FoxP3+ regulatory T cells (*Anderson et al., 2007*; *Jankovic et al., 2007*). However, experimental deletion of IL-10 production in T cells has been shown to promote the clearance of LCMV without any obvious collateral effects (*Clement et al., 2016*; *Parish et al., 2014*; *Richter et al., 2013*), suggesting a potential therapeutic role for similar manipulations in humans, albeit with the possibility of an attendant risk to the development of CD8+ T cell memory (*Laidlaw et al., 2015*).

The mechanisms that induce IL-10 expression in T cells require further clarification, despite proposed roles for costimulatory receptors, cytokines, transcription factors, and signals delivered via the T cell receptor (TCR) (*Saraiva and O'Garra, 2010*). For example, chronic antigen exposure and the transcription factor Blimp-1 appear to be important for the development of IL-10-producing CD4+ T cells in mice infected with LCMV (*Parish et al., 2014*), and the inhibitory receptor TIGIT is known to act upstream of IL-10 (*Schorer et al., 2020*). However, it is clear that viral persistence can be facilitated by IL-10, exemplified in the context of MCMV infection by ongoing replication in the salivary glands (SGs) (*Humphreys et al., 2007*; *Mandaric et al., 2012*).

Interferon (IFN)γ-expressing CD4+ T cells have been shown to limit viral replication in the SGs of mice infected with MCMV (*Jonjić et al., 1989*; *Lucin et al., 1992*; *Walton et al., 2011*). Nonetheless, CD4+ T cells also represent an important source of IL-10 (*Clement et al., 2016*; *Humphreys et al., 2007*), the production of which is promoted by IL-27 during acute infection with MCMV. In contrast, less is known about the mucosal IL-10-producing CD4+ T cells that appear during chronic infection with MCMV, which are phenotypically distinct from type 1 regulatory T (Tr1) cells, specifically lacking concurrent expression of CD49d and LAG-3, and develop independently of IL-27 (*Clement et al., 2016*). These cells express high levels of various transcription factors, such as c-Maf and T-bet (*Clement et al., 2016*), and often coexpress other molecules with putative inhibitory functions, such as PD-1, TIM-3, and IL-21 (*Apetoh et al., 2010*; *Awasthi et al., 2007*; *Chihara et al., 2018*; *Pot et al., 2009*; *Zhu et al., 2015*). However, the functional relevance of these characteristics has remained obscure, along with the ontogeny of IL-10-producing CD4+ T cells during chronic infection with MCMV.

To address these issues, we performed a comprehensive functional, phenotypic, and transcriptomic analysis of IL-10-producing CD4+ T cells isolated from the SGs of mice infected with MCMV. Our data revealed that these cells were clonally expanded and highly differentiated $T_H$1-like cells with gene expression signatures that indicated a key developmental role for T-bet. In addition, we identified an inhibitory effect attributable to arginase-1 (Arg1), which was upregulated among IL-10-producing CD4+ T cells during viral chronicity and facilitated the site-specific persistence of MCMV.

## Results

### IL-10-producing CD4+ T cells display a $T_H$1-like profile

To better understand the development and functionality of inhibitory CD4+ T cells that develop during viral chronicity, we infected MCMV IL-10 reporter (10BiT) mice with MCMV. These mice express Thy1.1 under the *Il10* promoter (*Maynard et al., 2007*). Unlike mucosal sites in the respiratory tract

(*Zhang et al., 2019*), ongoing viral replication in this model occurs primarily in the SGs, facilitated by the induction of CD4$^+$ T cells that produce IL-10 (*Humphreys et al., 2007*), which peak on day 14 post-infection (p.i.) (*Clement et al., 2016*). At this time point, we found that approximately 10–30% of CD4$^+$ T cells in the SGs were Thy1.1$^+$, of which ~95% displayed an effector memory phenotype (CD44$^{hi}$ CD62L$^{lo}$) (*Figure 1—figure supplement 1A*). IL-10-producing CD4$^+$ T cells were also induced by polyclonal stimulation and universally expressed Thy1.1 (*Figure 1—figure supplement 1B*).

We then compared the transcriptional profiles of endogenously generated IL-10$^+$ and IL-10$^-$ CD4$^+$ T cells, isolated via fluorescence-activated cell sorting (FACS) as Thy1.1$^+$ (IL-10$^+$) and Thy1.1$^-$ (IL-10$^-$) CD44$^{hi}$ CD62L$^{lo}$ CD4$^+$ T cells (*Figure 1—figure supplement 1A*). Principal component analysis (PCA) of the RNA-seq data revealed that Thy1.1$^+$ CD4$^+$ T cells were transcriptionally distinct from Thy1.1$^-$ CD4$^+$ T cells (*Figure 1—figure supplement 1C*). As expected, *Il10* was highly upregulated in Thy1.1$^+$ CD4$^+$ T cells (*Figure 1A*), and chromatin was more open in the *Il10* promoter region compared with Thy1.1$^-$ CD4$^+$ T cells (*Figure 1B*). Genes associated with localization and cell migration (*Ccl7*, *Cxcl2*, *Cxcl12*, *Ccl5*, *Cxcl14*, *Ccl28*, *Ccl12*, *Ccr1*, and *Ccr5*), cell signaling (*Ceacam1*, *Havcr2*, *Tigit*, *Lag3*, *Cd40*, *Cd36*, and *Itgb4*), regulation of cellular processes (*Prdm1*, *Gata2*, *Yes1*, *Card10*, and *Il33*), and metabolism (*Elovl7*, *Galnt3*, *Car13*, *Aldh1l1*, and *Ildrl*), including glycolysis and the tricarboxylic acid cycle (*Fbp2* and *Sdhc*), oxidative phosphorylation (*Osgin1*), and the mitochondrial respiratory chain (*Mt-Nd1*, *Ndufs6*, *Ndufb8*, *Uqcrfs1*, and *Uqcr11*), were also upregulated in Thy1.1$^+$ CD4$^+$ T cells, alongside genes associated with activation (*Fgl2*, *Cxcr2*, and *Nfil3*) and antiviral effector functions (*Gzmb*, *Prf1*, *Gzmk*, and *Lyz2*) (*Figure 1A, C, D* and *Figure 1—figure supplement 1D*). These latter gene profiles suggested the potential for cytolytic activity, but we found no evidence of a concomitant increase in the expression levels of granzyme B protein among Thy1.1$^+$ CD4$^+$ T cells (data not shown), which also lacked gene signatures classically associated with cytotoxic CD4$^+$ T cells, such as the upregulation of *Klrc1* and *Crtam* (https://doi.org/10.5281/zenodo.7243956).

Thy1.1$^+$ CD4$^+$ T cells are known to express the T$_H$1-associated chemokine receptors CXCR3 and CCR5 (*Clement et al., 2016*). We found that MCMV-induced Thy1.1$^+$ CD4$^+$ T cells shared many transcripts with CD4$^+$ T$_H$1 cells generated in vitro (*Stubbington et al., 2015*; *Figure 1E*), encompassing genes associated with numerous cellular and immunological processes (https://doi.org/10.5281/zenodo.7447477), and further expressed IFNγ in response to polyclonal stimulation at a population frequency of ~25% (*Figure 1—figure supplement 1E*). These data suggested that Thy1.1$^+$ CD4$^+$ T cells were commonly derived from antigen-specific T$_H$1 cells, especially given that IFNγ detection via flow cytometry likely underestimates the composite frequency of CD4$^+$ T cells that specifically recognize MCMV (*Jeitziner et al., 2013*). However, genes associated with the induction of IFNγ, including *Il18r1* and *il18rap*, were actually downregulated in Thy1.1$^+$ CD4$^+$ T cells (*Figure 1A, C, D*), and in two of three replicates, a similar pattern was observed for *Ifng* (https://doi.org/10.5281/zenodo.7243956). Comparable findings were reported previously in functional studies of IFNγ expression at the protein level among IL-10-producing CD4$^+$ T cells specific for HCMV or MCMV (*Clement et al., 2016*; *Mason et al., 2013*).

Other genes that were downregulated in Thy1.1$^+$ CD4$^+$ T cells included *Il7* and *Tcf1/7* (*Figure 1A, D*), which extended to the protein level (*Figure 1F*). The relative underexpression of these cell survival-associated factors coincided temporally with the rapid contraction of virus-specific IL-10-producing CD4$^+$ T cells that typically occurs during the early stages of viral chronicity (*Clement et al., 2016*). In contrast, Thy1.1$^+$ and Thy1.1$^-$ CD4$^+$ T cells expressed similar levels of transcripts encoding DR5 (https://doi.org/10.5281/zenodo.7243956), which engages natural killer (NK) cell-expressed TRAIL and induces CD4$^+$ T cell death in the SGs (*Schuster et al., 2014*).

IL-10 production among CD4$^+$ T cells has been associated with the expression of inhibitory molecules and markers of exhaustion (*Saraiva and O'Garra, 2010*). Counterintuitively, we found that MCMV-induced Thy1.1$^+$ CD4$^+$ T cells downregulated the exhaustion-associated transcription factor *Tox1* but nonetheless expressed a module of inhibitory genes, including *Lag3*, *Fgl2*, *Havcr2*, and *Entpd1* (*Figure 1D*). These inhibitory molecules were also expressed at the protein level, alongside PD-1 (*Figure 1G* and *Figure 1—figure supplement 1F*). In addition, differential bystander activation seemed unlikely, because Thy1.1$^+$ CD4$^+$ T cells expressed LAG-3 and PD-1 more commonly than Thy1.1$^-$ CD4$^+$ T cells after preselection based on the induction of IFNγ (*Figure 1—figure supplement 1E*).

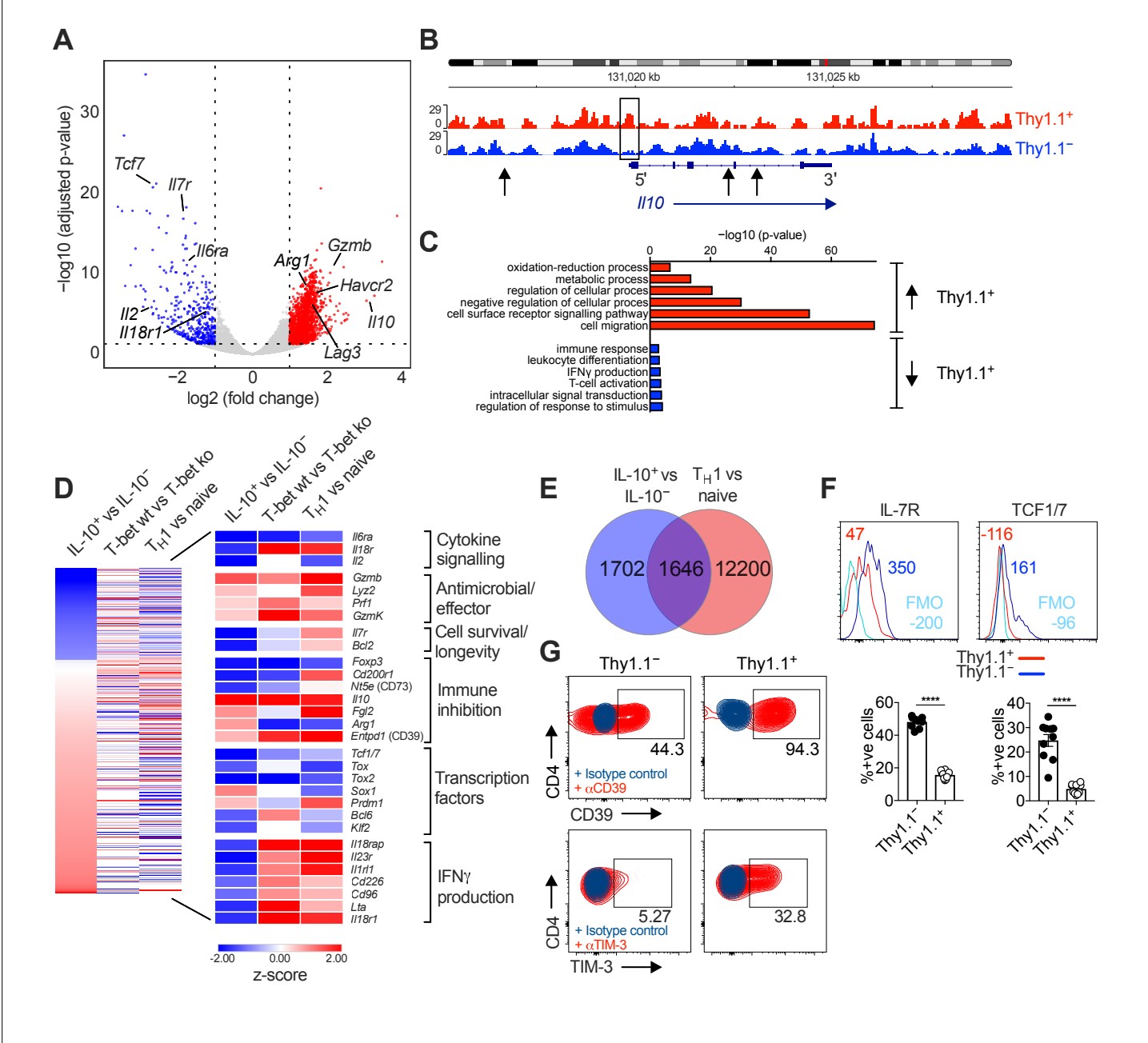

**Figure 1.** Interleukin (IL)-10-producing CD4+ T cells display a T_H1-like profile. 10BiT mice were infected with 3 × 10^4 pfu of murine cytomegalovirus (MCMV). Leukocytes were isolated from the salivary glands (SGs) on day 14 p.i. and sorted as CD4+ CD44+ CD62L− CD90/90.1+ (Thy1.1+) or CD90/90.1− (Thy1.1−) populations via fluorescence-activated cell sorting (FACS). (**A**) Volcano plot highlighting differentially upregulated genes in Thy1.1+ CD4+ T cells (red) versus Thy1.1− CD4+ T cells (blue). (**B**) ATAC-seq profiles showing accessible chromatin regions in the *Il10* gene for Thy1.1+ CD4+ T cells (red) and Thy1.1− CD4+ T cells (blue). Data are shown as normalized values accounting for the total number of reads per lane. The black box indicates a major difference in chromatin accessibility. Black arrows indicate binding motifs for Tbx21. (**C**) Gene ontology analysis of data from (**A**) indicating the top six modules that were upregulated (red) or downregulated (blue) in Thy1.1+ CD4+ T cells. (**D**) Heatmap comparing data from (**A**) (left column) with published data from T-bet+ versus T-bet-knockout CD4+ T cells (middle column, GSE38808) and T_H1 versus naive CD4+ T cells (right column, E-MTAB-2582). Displayed genes were selected according to relevant pathways identified via gene ontology analysis and tabulated against respective functions (all p < 0.05). (**E**) Venn diagram showing the overlap between genes enriched in Thy1.1+ CD4+ T cells (**A, D**) and genes enriched in T_H1-like CD4+ T cells (E-MTAB-2582). Data in (**A–E**) are shown as pooled analyses from a minimum of n = 5 mice per group representing three independent experiments. (**F**) Representative histograms (top) and summary bar graphs (bottom) showing the expression of IL-7R and TCF1/7 among Thy1.1+ CD4+ T cells (red) and Thy1.1− CD4+ T cells (blue). The fluorescence-minus-one control is shown in sky blue (top). Bottom: data are shown as mean ± standard error of the mean (SEM; n = 10 mice per group representing two independent experiments). ****p < 0.0001 (Mann–Whitney U test). (**G**) Representative flow

*Figure 1 continued on next page*

# eLife Research article

Immunology and Inflammation | Microbiology and Infectious Disease

Collectively, these data showed that IL-10-producing CD4⁺ T cells exhibited a highly differentiated T$_H$1-like profile, characterized by the upregulation of various inhibitory molecules and the downmodulation of IFNγ expression lacking concordance with known signatures of exhaustion, during chronic infection with MCMV.

## IL-10-producing CD4⁺ T cells exhibit prominent clonal structures

IL-10-producing CD4⁺ T cells recognize a broad range of viral antigens during chronic infection with MCMV (*Clement et al., 2016*). To characterize these interactions in more detail and evaluate the clonal relationship between IL-10⁺ (Thy1.1⁺) and IL-10⁻ (Thy1.1⁻) CD4⁺ T cells, we used a next-generation approach to sequence the corresponding TCRs.

The repertoires of Thy1.1⁺ CD4⁺ T cells were less diverse and incorporated more prominent clonal expansions compared with the repertoires of Thy1.1⁻ CD4⁺ T cells (*Figure 2A, B* and *Figure 2—figure supplement 1A*). Several features also indicated that these expansions represented antigen-focused responses confined largely to Thy1.1⁺ CD4⁺ T cells (*Figure 2C–G*). First, the number of nucleotide variants that encoded each complementarity-determining region (CDR)3α and CDR3β amino acid sequence, an indicator of antigen-specific convergence (*Logunova et al., 2020*), was higher overall among Thy1.1⁺ CD4⁺ T cells versus Thy1.1⁻ CD4⁺ T cells (*Figure 2C*). Second, there were some differences in *Trbv* gene use that distinguished Thy1.1⁺ CD4⁺ T cells from Thy1.1⁻ CD4⁺ T cells, albeit with a general preference for *Trbv3*, *Trbv5*, and *Trbv31* (*Figure 2D*). Third, clusters of homologous TCRβ variants, identified using the statistical model ALICE (*Pogorelyy et al., 2019*), were detected predominantly among Thy1.1⁺ CD4⁺ T cells (*Figure 2E–G*). Importantly, this latter model accounts for generation probabilities, reliably separating immunologically relevant and irrelevant public TCRs.

It should be noted that none of these differences were absolute. For example, the clusters of TCRβ variants identified among Thy1.1⁺ CD4⁺ T cells also occurred at lower cumulative frequencies among Thy1.1⁻ CD4⁺ T cells (*Figure 2E–G*), and the TCRα and TCRβ repertoires overlapped considerably between Thy1.1⁺ CD4⁺ T cells and Thy1.1⁻ CD4⁺ T cells (*Figure 2—figure supplement 1B*). Moreover, there were no prominent differences in the physicochemical properties of amino acids in the central parts of the CDR3α and CDR3β loops, which generally differ among functionally discrete subsets of CD4⁺ T cells (*Kasatskaya et al., 2020*), to indicate an ontogenetic divergence between Thy1.1⁺ CD4⁺ T cells and Thy1.1⁻ CD4⁺ T cells (*Figure 2—figure supplement 1C*).

Collectively, these data revealed the presence of common molecular signatures that predominated among Thy1.1⁺ CD4⁺ T cells, consistent with the notion of an antigen-driven process of differentiation leading to the production of IL-10.

## IL-10-producing CD4⁺ T cells are enriched for expression of Arg1

Our analysis of inhibitory gene expression revealed one particularly intriguing feature, namely that Thy1.1⁺ CD4⁺ T cells significantly upregulated *Arg1* (*Figure 1D*). Arg1 promotes the catalytic breakdown of L-arginine (*Munder, 2009*) and has been shown to inhibit the proliferation of T cells (*Czystowska-Kuzmicz et al., 2019*; *Rodriguez et al., 2004*; *Rodriguez et al., 2002*). A previous study also reported that T cells could express *Arg1* (*Washburn et al., 2019*), although the functional relevance of this observation has remained obscure.

To address this knowledge gap, we first confirmed expression at the protein level via Western blotting (*Figure 3A*) in experiments incorporating control mice lacking the ability to express Arg1 in the CD4⁺ lineage (*Cd4^{Cre/+}Arg1^{flox/flox}*). We then revealed the open chromatin structure around *Arg1* in Thy1.1⁺ CD4⁺ T cells (*Figure 3B*) and further probed the expression of Arg1 versus Thy1.1 among

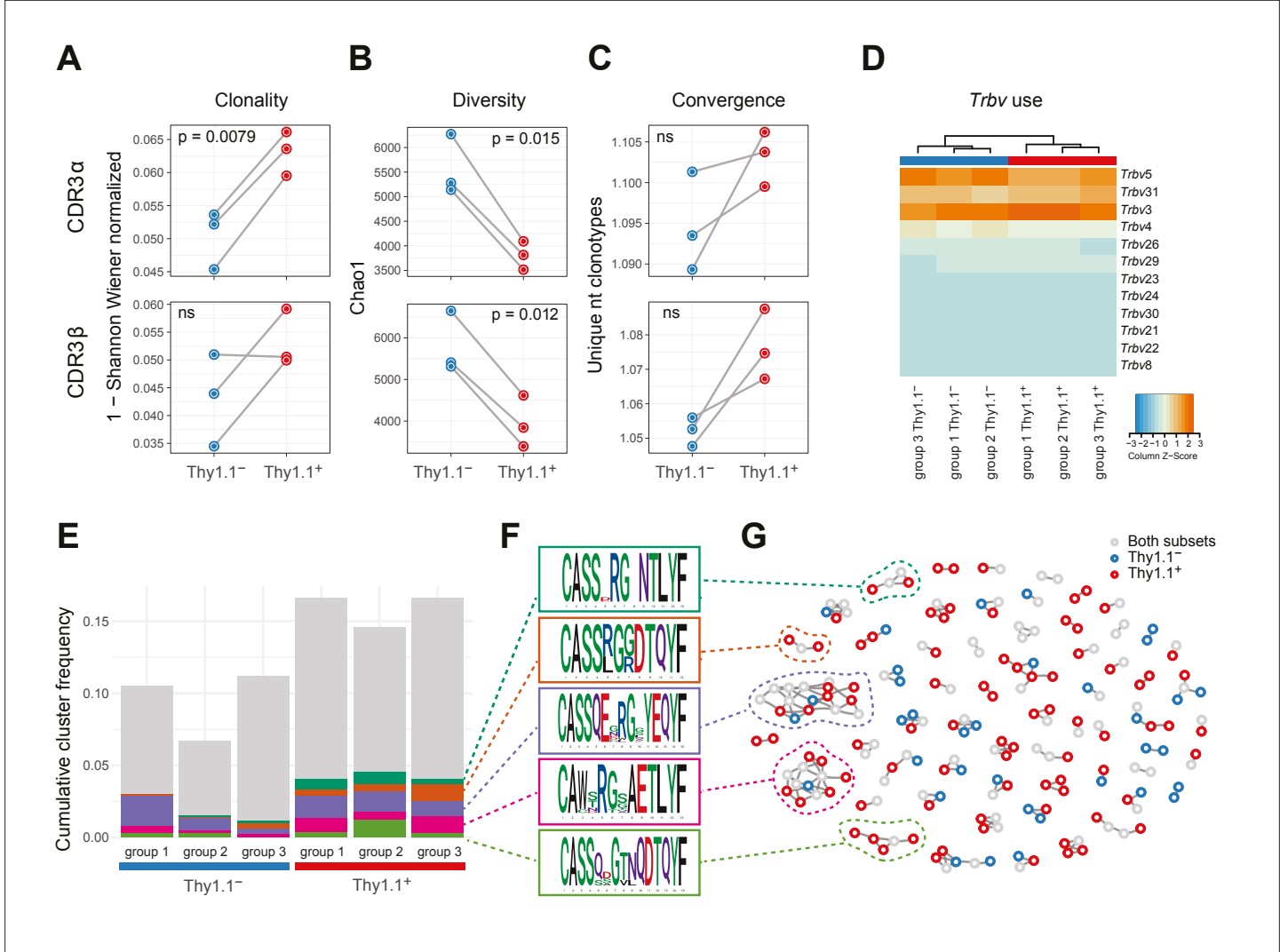

**Figure 2.** Interleukin (IL)-10-producing CD4+ T cells exhibit prominent clonal structures. 10BiT mice were infected with 3 × 10⁴ pfu of murine cytomegalovirus (MCMV). Leukocytes were isolated from the salivary glands (SGs) on day 14 p.i. and sorted as CD4+ CD44+ CD62L⁻ CD90/90.1+ (Thy1.1+) or CD90/90.1⁻ (Thy1.1⁻) populations via fluorescence-activated cell sorting (FACS). (**A**) Clonality and (**B**) diversity metrics calculated for the T cell receptor (TCR)α (top) and TCRβ repertoires (bottom) derived from Thy1.1+ CD4+ T cells and Thy1.1⁻ CD4+ T cells. (**C**) TCR convergence measured as the average number of nucleotide sequences encoding amino acid-identical complementarity-determining region (CDR)3α (top) and CDR3β loops (bottom) across the 2000 most prevalent clonotypes. (**A–C**) p values were calculated using a paired t-test with Benjamini–Hochberg correction. ns, not significant. (**D**) Hierarchical clustering of *Trbv* gene use weighted by clonotype frequency. (**E–G**) Cluster analysis of the 500 most prevalent TCRβ clonotypes using the tcrgrapher pipeline. (**E**) The cumulative frequency of tcrgrapher hits per sample is shown in gray. The frequency of each cluster comprising at least two tcrgrapher hits was calculated for each sample and averaged across all six repertoires. The five most prevalent clusters are shown in color. (**F**) Amino acid logos for each of the five most prevalent clusters. (**G**) Visual representation of clusters comprising at least two tcrgrapher hits. Nodes represent unique amino acid sequences. Edges connect sequences with a single amino acid mismatch. Amino acid sequences present only in Thy1.1+ CD4+ T cells are shown in red, amino acid sequences present only in Thy1.1⁻ CD4+ T cells are shown in blue, and amino acid sequences present in both Thy1.1+ CD4+ T cells and Thy1.1⁻ CD4+ T cells are shown in gray. Data are shown as pooled analyses from n = 4 mice per group representing three independent experiments (groups 1–3).

The online version of this article includes the following figure supplement(s) for figure 2:

**Figure supplement 1.** Analysis of repertoire diversity and overlap and the physicochemical properties of T cell receptors (TCRs).

CD4+ T cells via flow cytometry (*Figure 3C*). Our data showed that Arg1 was expressed by CD4+ T cells almost exclusively in the SGs (*Figure 3A, C*). Of note, Arg1 expression was also detected among CD8+ T cells but not among NK T cells via flow cytometry, although intracellular discrimination was subtle (*Figure 3—figure supplement 1A*). Depletion experiments nonetheless revealed that only CD4+ T

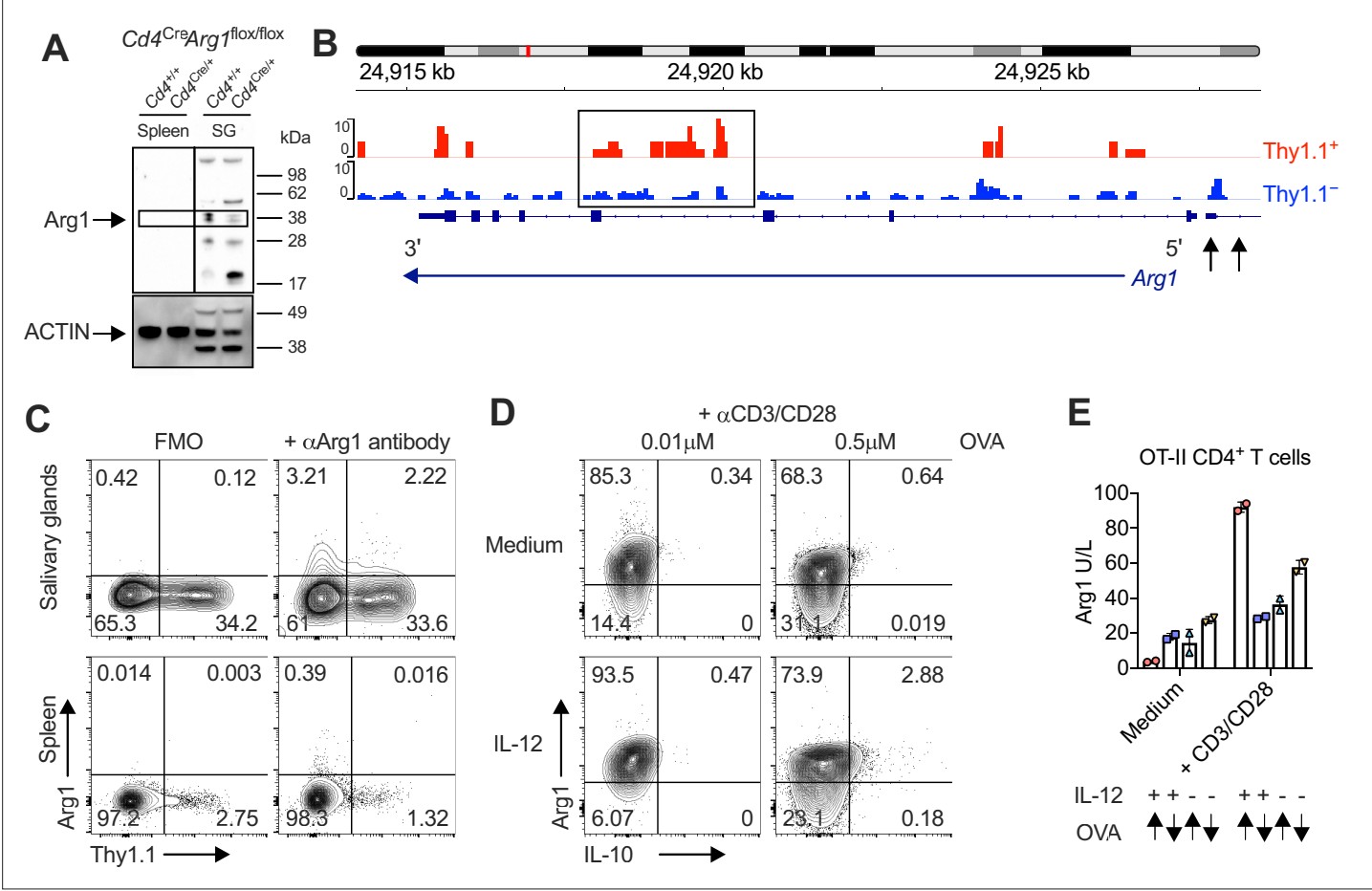

**Figure 3.** Interleukin (IL)-10-producing CD4⁺ T cells are enriched for expression of arginase-1 (Arg1). (**A**) Expression of Arg1 among leukocytes isolated via magnetic separation from the salivary glands (SGs) and spleens of $Cd4^{+/+}Arg1^{flox/flox}$ or $Cd4^{Cre/+}Arg1^{flox/flox}$ mice on day 14 p.i. detected by Western blot. (**B, C**) 10BiT mice were infected with $3 \times 10^4$ pfu of murine cytomegalovirus (MCMV). (**B**) Leukocytes were isolated from the SGs on day 14 p.i. and sorted as CD4⁺ CD44⁺ CD62L⁻ CD90/90.1⁺ (Thy1.1⁺) or CD90/90.1⁻ (Thy1.1⁻) populations via fluorescence-activated cell sorting (FACS). ATAC-seq profiles show accessible chromatin regions in the $Arg1$ gene for Thy1.1⁺ CD4⁺ T cells (red) and Thy1.1⁻ CD4⁺ T cells (blue). Data are shown as normalized values accounting for the total number of reads per lane. The black box indicates a major difference in chromatin accessibility. Black arrows indicate binding motifs for Tbx21. Data are shown as pooled analyses from a minimum of $n$ = 5 mice per group representing three independent experiments. (**C**) Representative flow cytometry plots showing the expression of Arg1 versus Thy1.1 among CD4⁺ T cells isolated from the SGs and spleens on day 14 p.i. (**A, C**) Data are shown as pooled analyses from a minimum of $n$ = 7 mice per group representing three independent experiments. (**D**) Representative flow cytometry plots showing the expression of Arg1 versus IL-10 among OT-II-specific CD4⁺ T cells generated in vitro in the absence or presence of IL-12 (3 ng/ml) ± OVA₃₂₃₋₃₃₉ for 7 days and then stimulated with anti-CD3/CD28 for 4 hr. (**E**) Summary bar graph showing Arg1 protein concentrations in culture supernatants from (**D**) after stimulation with anti-CD3/CD28 for 48 hr. Up arrows indicate the higher concentration of OVA₃₂₃₋₃₃₉ (0.5 μM), and down arrows indicate the lower concentration of OVA₃₂₃₋₃₃₉ (0.01 μM). Data are shown as mean ± standard error of the mean (SEM).

The online version of this article includes the following source data and figure supplement(s) for figure 3:

**Source data 1.** Interleukin (IL)-10-producing CD4⁺ T cells are enriched for expression of arginase-1 (Arg1).

**Source data 2.** Interleukin (IL)-10-producing CD4⁺ T cells are enriched for expression of arginase-1 (Arg1).

**Figure supplement 1.** T cell-specific deletion of arginase-1 (Arg1) does not impact the function or phenotype of T cells in naive mice.

**Figure supplement 1—source data 1.** T cell-specific deletion of arginase-1 (Arg1) does not impact the function or phenotype of T cells in naive mice.

cells contributed significantly to Arg1 protein concentrations in SG homogenates during chronic infection with MCMV (*Figure 3—figure supplement 1B*).

IL-10-producing T_H1 cells can be induced experimentally via high-dose antigen stimulation in the presence of IL-12 (*Saraiva et al., 2009*). Accordingly, we hypothesized that Arg1⁺ IL-10-producing CD4⁺ T cells might develop under similar conditions in vitro, given the corresponding T_H1-like profile observed during chronic infection with MCMV. To test this notion, we stimulated ovalbumin

(OVA)-specific transgenic CD4+ T cells from OT-II mice with high or low doses of the cognate peptide (OVA$_{323-339}$) in the absence or presence of IL-12. Despite constitutively high expression levels of Arg1, only high-dose OVA$_{323-339}$ in combination with IL-12 induced the development of CD4+ T cells that expressed Arg1 and IL-10, and importantly, all IL-10-producing CD4+ T cells coexpressed Arg1 (*Figure 3D*). Arg1 protein concentrations in culture supernatants also increased substantially after stimulation with high-dose OVA$_{323-339}$ in the presence of IL-12 (*Figure 3E*).

Collectively, these data suggested that Arg1 expression was a hallmark of IL-10-producing CD4+ T cells, which developed almost exclusively in the SGs during chronic infection with MCMV.

## CD4+ T cells promote viral persistence via expression of Arg1

To explore the biological relevance of our findings, we infected *Cd4*$^{+/+}$*Arg1*$^{flox/flox}$ and *Cd4*$^{Cre/+}$*Arg1*$^{flox/flox}$ mice with MCMV. Lineage-specific deletion of *Arg1* did not impact the function or phenotype of CD4+ T cells in naive mice (*Figure 3—figure supplement 1C, D*). Higher numbers of IFNγ-expressing CD4+ T cells (*Figure 4A*) and higher frequencies of proliferating (Ki-67+) CD4+ T cells (*Figure 4B*) were nonetheless observed after viral antigen stimulation in the SGs of *Cd4*$^{Cre/+}$*Arg1*$^{flox/flox}$ versus *Cd4*$^{+/+}$*Arg1*$^{flox/flox}$ mice during the chronic phase of infection with MCMV. These data suggested that Arg1 inhibited the proliferation of CD4+ T cells in vivo, consistent with a previous in vitro study (*Munder et al., 2006*). Similarly, higher numbers of virus-specific CD8+ T cells, quantified using tetrameric antigen probes, were detected in the spleens of *Cd4*$^{Cre/+}$*Arg1*$^{flox/flox}$ versus *Cd4*$^{+/+}$*Arg1*$^{flox/flox}$ mice on day 30 p.i. (*Figure 4C*). In contrast, *Arg1* deletion had no significant impact on the development of virus-specific IL-10-producing CD4+ T cells in the SGs (*Figure 4D*).

IFNγ-expressing CD4+ T cells are known to restrict MCMV replication in the SGs (*Walton et al., 2011*). Accordingly, we found that viral shedding in the saliva (*Figure 4E*) and viral replication in the SGs (*Figure 4F*) were reduced in *Cd4*$^{Cre/+}$*Arg1*$^{flox/flox}$ versus *Cd4*$^{+/+}$*Arg1*$^{flox/flox}$ mice during the chronic phase of infection with MCMV. Importantly, no differences in viral replication were observed between *Cd4*$^{Cre/+}$*Arg1*$^{flox/flox}$ and control mice at an earlier time point (day 14 p.i.) (*Figure 4—figure supplement 1A*), and deletion of *Arg1* in myeloid cells, achieved using *Lyz2*$^{Cre/+}$*Arg1*$^{flox/flox}$ mice, had no impact on viral shedding in the saliva or viral replication in the SGs (*Figure 4—figure supplement 1B, C*). Of note, there was also no evidence that *Cd4*$^{Cre/+}$*Arg1*$^{flox/flox}$ mice were more susceptible to virus-induced autoimmunity, as indicated by anti-Sjögrens syndrome antigen (SSA) titers comparable to those observed in *Cd4*$^{+/+}$*Arg1*$^{flox/flox}$ mice (*Figure 4—figure supplement 1D*).

Collectively, these data indicated that Arg1 expression by CD4+ T cells selectively inhibited the accumulation of virus-specific CD4+ and CD8+ T cells in the SGs, leading to suboptimal immune control of viral replication during chronic infection with MCMV.

## IL-10-producing CD4+ T cells develop in a T-bet-dependent manner

In line with our previous work (*Clement et al., 2016*), we noted that Thy1.1+ CD4+ T cells expressed higher amounts of T-bet at the protein level than Thy1.1− CD4+ T cells (*Figure 5A*), and concordantly, we found that open chromatin was enriched in the *Tbx21* region of Thy1.1+ CD4+ T cells versus Thy1.1− CD4+ T cells (*Figure 5B*). No such differences in expression intensity were observed for the related T-box transcription factor Eomesodermin (Eomes) (*Figure 5—figure supplement 1A, B*), which promotes the development of Tr1 cells (*Roessner et al., 2021*; *Zhang et al., 2017*). We also detected considerable overlap between the gene expression profiles of Thy1.1+ CD4+ T cells and T-bet-orchestrated CD4+ T$_H$1 cells generated in vitro (*Zhu et al., 2012*; *Figure 5C*). Moreover, T-bet suppresses the expression of TCF1/7 and IL-7R (*Dominguez et al., 2015*; *Oestreich et al., 2011*), mimicking key phenotypic features of Thy1.1+ CD4+ T cells (*Figure 1D, F*). On the basis of these observations, we hypothesized that T-bet promoted the development of Arg1+ IL-10-producing CD4+ T cells during chronic infection with MCMV.

To test this notion, we crossed tamoxifen-inducible *Cd4*$^{CreERT2/+}$ mice with *Tbx21*$^{flox/flox}$ mice (*Intlekofer et al., 2008*), allowing us to suppress T-bet expression at the onset of viral chronicity (day 7 p.i.). These mice were further crossed to incorporate a *Rosa26*$^{flSTOPtdRFP}$ allele (*Luche et al., 2007*), enabling the identification of cells in which *Cre* was expressed via the detection of tandem-dimer red fluorescent protein (tdRFP). Mice were then infected with MCMV. Tamoxifen was administered daily for 5 days from day 7 p.i. to deplete T-bet after initial antigen exposure in a CD4-dependent manner, an

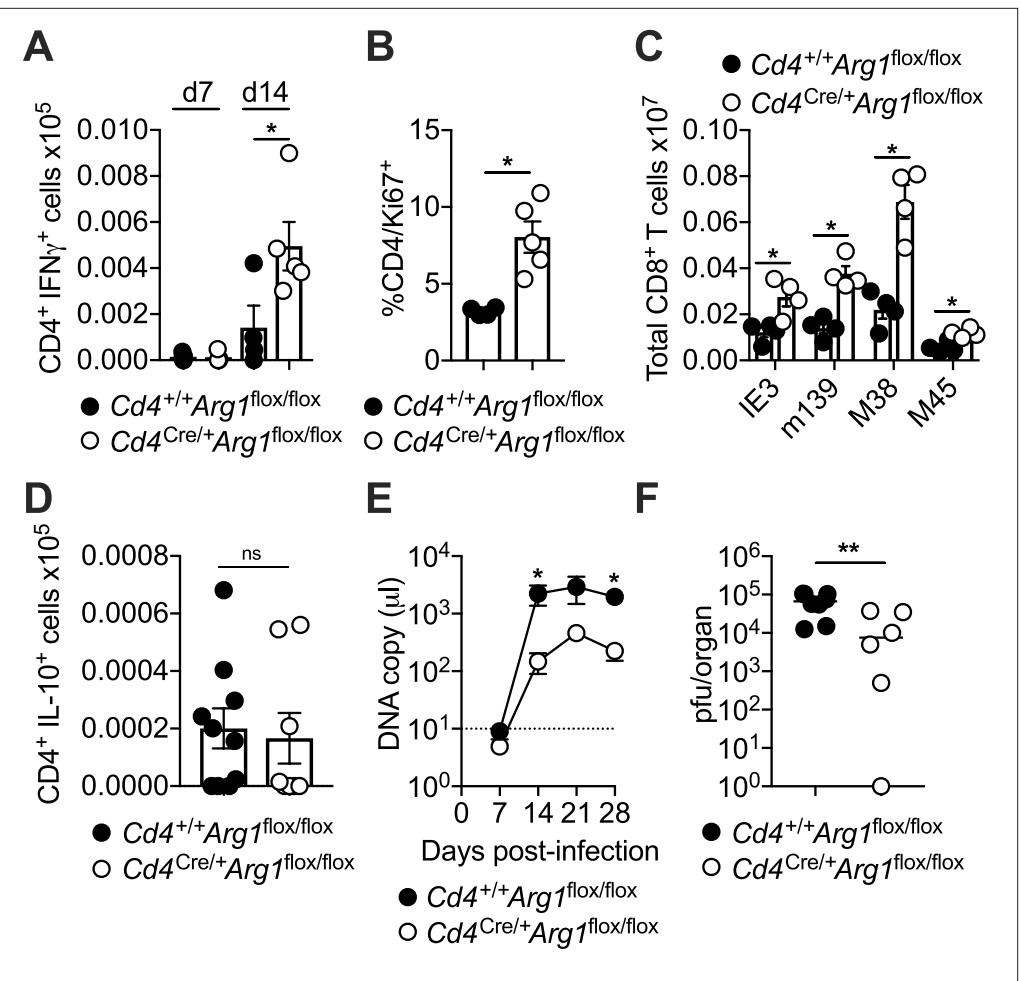

**Figure 4.** CD4[+] T cells promote viral persistence via expression of arginase-1 (Arg1). *Cd4*[+/+]*Arg1*[flox/flox] and *Cd4*[Cre/+]*Arg1*[flox/flox] mice were infected with 3 × 10[4] pfu of murine cytomegalovirus (MCMV). (**A**) MCMV-specific CD4[+] T cell responses in the salivary glands (SGs) on days 7 and 14 p.i. measured using flow cytometry to detect interferon (IFN)γ. Immunodominant peptides were pooled for stimulation. Data are shown as mean ± standard error of the mean (SEM; *n* = 4–6 mice per group representing three independent experiments). (**B**) Expression of Ki-67 among CD4[+] T cells isolated from the SGs on day 14 p.i. measured via flow cytometry. Data are shown as mean ± SEM (*n* = 4–5 mice per group representing two independent experiments). (**C**) MCMV tetramer[+] CD8[+] T cells quantified in spleens on day 30 p.i. via flow cytometry. Data are shown as mean ± SEM (*n* = 4 mice per group representing two independent experiments). (**D**) MCMV-specific CD4[+] T cell responses in the SGs on days 7 and 14 p.i. measured using flow cytometry to detect interleukin (IL)-10. Immunodominant peptides were pooled for stimulation. Data are shown as mean ± SEM (*n* = 8–10 mice per group representing two independent experiments). (**A–D**) *p < 0.05 (Mann–Whitney *U* test). (**E**) Viral genomes in saliva on days 7, 14, 21, and 28 p.i. measured via qPCR. Data are shown as mean ± SEM (*n* = 8 mice per group representing two independent experiments). *p < 0.05 (Mann–Whitney *U* test). (**F**) MCMV replication in SG homogenates on day 30 p.i. measured via plaque assay. Data are shown as individual points with median values (*n* = 6–7 mice per group representing two or three independent experiments). **p < 0.01 (Mann–Whitney *U* test).

The online version of this article includes the following source data and figure supplement(s) for figure 4:

**Source data 1.** CD4[+] T cells promote viral persistence via expression of arginase-1 (Arg1).

**Figure supplement 1.** Arginase-1 (Arg1) expression in myeloid cells does not impact the replication of murine cytomegalovirus (MCMV).

**Figure supplement 1—source data 1.** Arginase-1 (Arg1) expression in myeloid cells does not impact the replication of murine cytomegalovirus (MCMV).

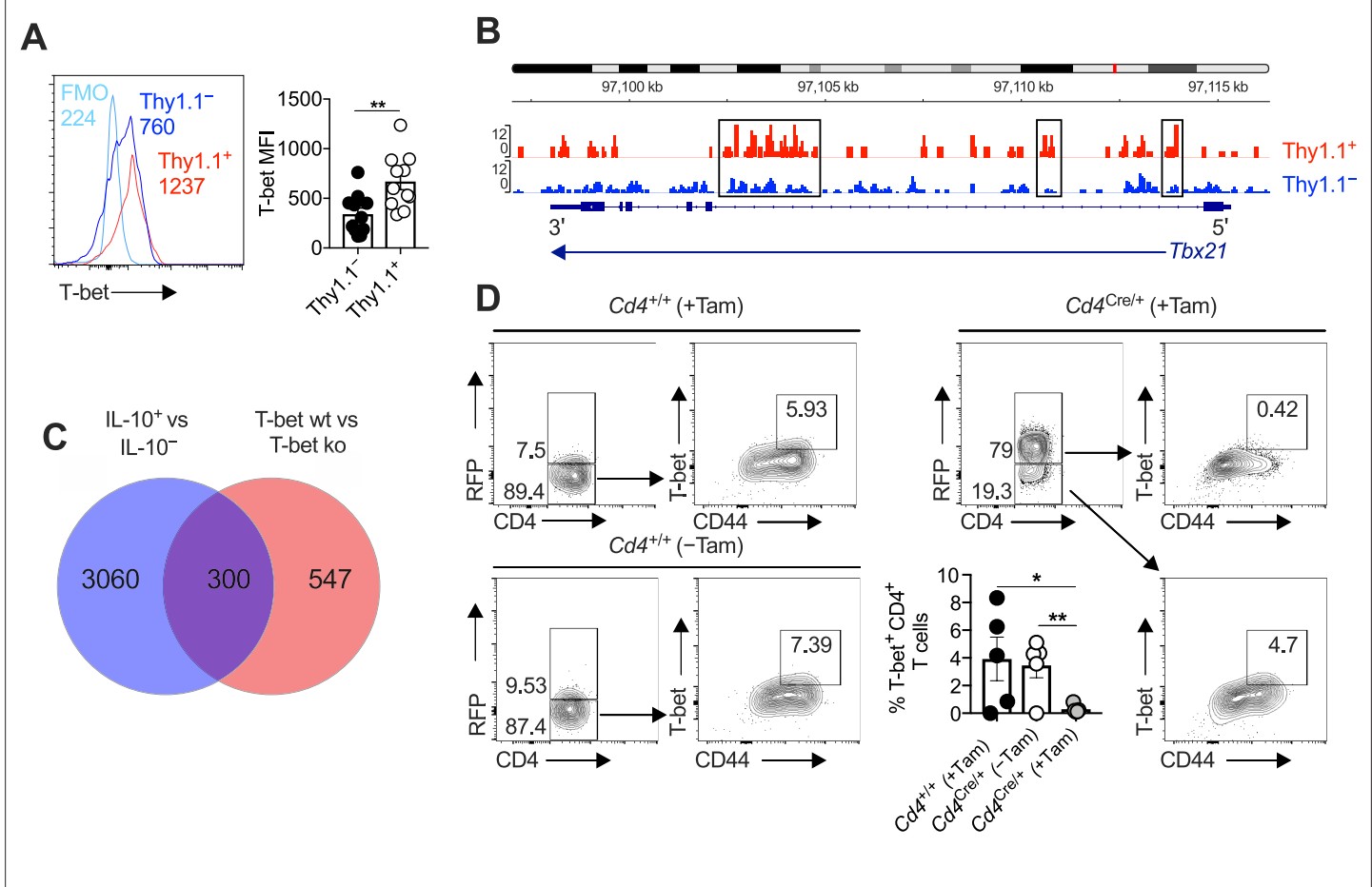

**Figure 5.** Interleukin (IL)-10-producing CD4+ T cells express T-bet and T-bet-inducible genes. (**A, B**) 10BiT mice were infected with 3 × 10⁴ pfu of murine cytomegalovirus (MCMV). (**A**) Leukocytes were isolated from the salivary glands (SGs) on day 14 p.i. Representative histograms (left) and summary bar graph (right) show the expression of T-bet among Thy1.1+ CD4+ T cells (red) and Thy1.1− CD4+ T cells (blue). Data are shown as mean ± standard error of the mean (SEM; *n* = 5–6 mice per group pooled from two independent experiments). MFI, median fluorescence intensity. **p < 0.01 (Mann–Whitney *U* test). (**B**) Leukocytes were isolated from the SGs on day 14 p.i. and sorted as CD4+ CD44+ CD62L− CD90/90.1+ (Thy1.1+) or CD90/90.1− (Thy1.1−) populations via fluorescence-activated cell sorting (FACS). ATAC-seq profiles show accessible chromatin regions in the *Tbx21* gene for Thy1.1+ CD4+ T cells (red) and Thy1.1− CD4+ T cells (blue). Data are shown as normalized values accounting for the total number of reads per lane. The black boxes indicate major differences in chromatin accessibility. Data are shown as pooled analyses from a minimum of *n* = 5 mice per group representing three independent experiments. (**C**) Venn diagram showing the overlap between genes enriched in Thy1.1+ CD4+ T cells (**A, D**) and genes enriched in T-bet+ CD4+ T cells (GSE38808). (**D**) *Cd4+/+Tbx21*flox/flox (*Cd4+/+*) and *Cd4*CreERT2/+*Tbx21*flox/flox (*Cd4*Cre/+*) mice were infected with 3 × 10⁴ pfu of MCMV. Tamoxifen was administered (+Tam) or withheld (−Tam) from days 7 to 12 p.i. Leukocytes were isolated from the SGs on day 14 p.i. Representative flow cytometry plots show the expression of CD4 versus RFP (left) and CD44 versus T-bet for the gated populations (right). Data in the bar graph are shown as mean ± SEM (*n* = 4–5 mice per group representing four or five independent experiments). *p < 0.05, **p < 0.01 (Mann–Whitney *U* test).

The online version of this article includes the following source data and figure supplement(s) for figure 5:

**Source data 1.** Interleukin (IL)-10-producing CD4+ T cells express T-bet and T-bet-inducible genes.

**Figure supplement 1.** Interleukin (IL)-10-producing CD4+ T cells are not selectively regulated by Eomes.

**Figure supplement 1—source data 1.** Interleukin (IL)-10-producing CD4+ T cells are not selectively regulated by Eomes.

effect that clearly associated with the coincident expression of RFP (*Figure 5D* and *Figure 5—figure supplement 1C*).

CD4-specific T-bet depletion reduced the accumulation of virus-specific IL-10-producing CD4+ T cells (*Figure 6A*) and Arg1+ CD4+ T cells by day 14 p.i. (*Figure 6B, C*). However, it seemed unlikely that T-bet directly stimulated the expression of IL-10 and Arg1, because the corresponding binding motifs were not preferentially accessible in the *Il10* and *Arg1* regions (*Figure 1B* and *Figure 3B*). Instead, we observed a shift toward a less differentiated phenotype in *Cd4*CreERT2/+*Tbx21*flox/flox mice (*Figure 6D–G*),

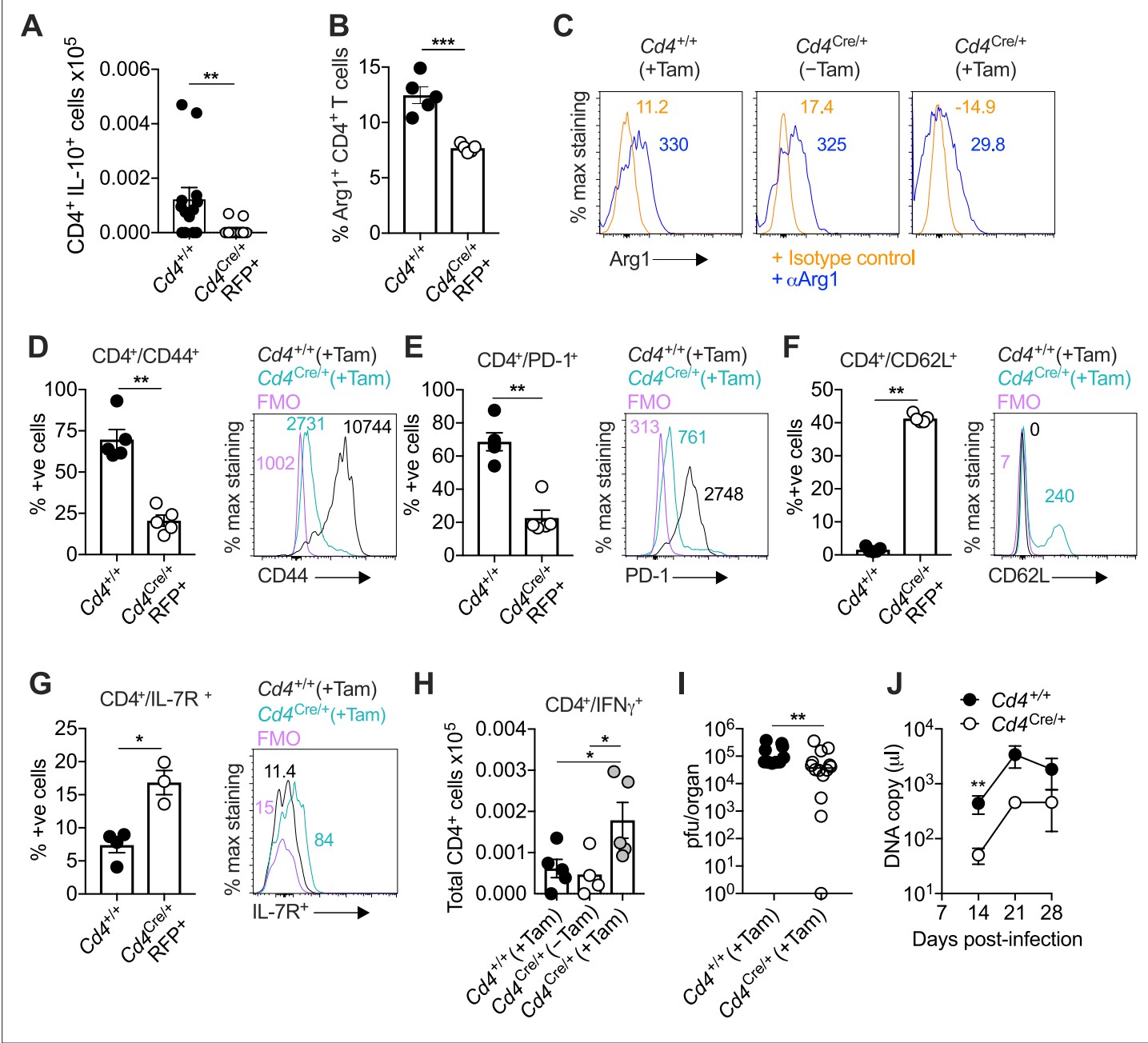

**Figure 6.** Interleukin (IL)-10-producing CD4⁺ T cells develop in a T-bet-dependent manner. (**A–J**) $Cd4^{+/+}Tbx21^{flox/flox}$ ($Cd4^{+/+}$) and $Cd4^{CreERT2/+}Tbx21^{flox/flox}$ ($Cd4^{Cre/+}$) mice were infected with $3 \times 10^4$ pfu of murine cytomegalovirus (MCMV). Tamoxifen was administered (+Tam) or withheld (−Tam) from days 7 to 12 p.i. Leukocytes were isolated from the salivary glands (SGs) on day 14 p.i. (**A**) MCMV-specific CD4⁺ T cell responses in the SGs measured using flow cytometry to detect IL-10. Immunodominant peptides were pooled for stimulation. Data are shown as mean ± standard error of the mean (SEM; $n$ = 4–5 mice per group representing two independent experiments). **p < 0.01 (Mann–Whitney $U$ test). Summary bar graph (**B**) and representative histograms (**C**) showing the expression of arginase-1 (Arg1) among CD4⁺ T cells measured via flow cytometry. Data are shown as mean ± SEM ($n$ = 5 mice per group representing two independent experiments). ***p < 0.001 (Mann–Whitney $U$ test). Summary bar graphs (left) and representative histograms (right) showing the expression of CD44 (**D**), PD-1 (**E**), CD62L (**F**), and IL-7R (**G**) among CD4⁺ T cells measured via flow cytometry. Data in (**D–F**) are shown as mean ± SEM ($n$ = 5 mice per group representing two independent experiments). Data in (**G**) are shown as mean ± SEM ($n$ = 3–4 mice per group representing two independent experiments). *p < 0.05, **p < 0.01 (Mann–Whitney $U$ test). (**H**) MCMV-specific CD4⁺ T cell responses in the SGs measured using flow cytometry to detect interferon (IFN)γ. Immunodominant peptides were pooled for stimulation. Data are shown as mean ± SEM ($n$ = 4–5 mice per group representing two independent experiments). *p < 0.05 (Mann–Whitney $U$ test). (**I**) MCMV replication in SG homogenates on day 14 p.i. measured via plaque assay. Data are shown as individual points with median values ($n$ = 10–14 mice per group pooled from three independent experiments). **p < 0.01 (Mann–Whitney $U$ test). (**J**) Viral genomes in saliva on days 7, 14, 21, and 28 p.i. measured via qPCR. Data are shown as mean

*Figure 6 continued on next page*

*Figure 6 continued*

± SEM (*n* = 5 mice per group representing two independent experiments). **p < 0.01 (Mann–Whitney *U* test). Data in (**A–I**) show all groups after the administration of tamoxifen.

The online version of this article includes the following source data for figure 6:

**Source data 1.** Interleukin (IL)-10-producing CD4+ T cells develop in a T-bet-dependent manner.

with decreased expression of CD44 (*Figure 6D*) and PD-1 (*Figure 6E*) and increased expression of CD62L (*Figure 6F*) and IL-7R (*Figure 6G*) in the absence of T-bet. In addition, T-bet depletion was associated with increased numbers of virus-specific IFNγ-expressing CD4+ T cells (*Figure 6H*) and enhanced control of viral replication in the SGs (*Figure 6I*), as well as reduced salivary shedding of MCMV (*Figure 6J*). In accordance with the observation that IL-10-producing CD4+ T cells occur transiently during the early stages of viral chronicity (*Clement et al., 2016*), we also found that CD4-specific T-bet depletion was no longer protective by day 28 p.i. (*Figure 6J* and *Figure 5—figure supplement 1D*).

Collectively, these results suggested that T-bet exhibited dual functionality under conditions of recurrent antigen stimulation, concurrently driving the accumulation of virus-specific $T_H$1 responses and promoting the development of Arg1+ IL-10-producing CD4+ T cells, which together shaped immune control of viral replication during the early stages of chronic infection with MCMV.

## Discussion

In this study, we identified IL-10-producing CD4+ T cells as highly differentiated $T_H$1-like cells, which occurred transiently at mucosal sites of viral persistence under the strict governance of T-bet. These cells also expressed the metabolic enzyme Arg1. Further investigation revealed that Arg1 expression was a critical immune regulatory function of CD4+ T cells, suppressing antiviral immunity and facilitating viral chronicity in the context of infection with MCMV.

Our finding that *Arg1* was one of the putative immune regulatory genes expressed by IL-10-producing CD4+ T cells was initially counterintuitive, given the reported lack of Arg1 protein expression by human CD4+ T cells in vitro (*Geiger et al., 2016*). However, *Arg1* expression by CD4+ and CD8+ T cells has been described in patients with sepsis (*Washburn et al., 2019*), indicating a potential role for generic infectious stimuli and/or an antigen-driven process mediated via the TCR. The latter possibility is consistent with our repertoire data, which revealed that prominent clonal expansions were associated with the production of IL-10. Importantly, we demonstrated that Arg1 was expressed at the protein level, both in secreted form and intracellularly. It should be noted that relatively weak intracellular signals were detected in flow cytometric analyses of C57BL/6 mice, akin to an earlier study of patients infected with HBV (*Pallett et al., 2015*). Western blotting experiments nonetheless clearly showed that CD4+ T cells expressed Arg1 in the SGs of mice chronically infected with MCMV. Our flow cytometry experiments also suggested that CD8+ T cells could express Arg1, albeit with the caveat of suboptimal visualization, but importantly, CD4+ T cells secreted far greater amounts of soluble Arg1, at least in the milieu of the SGs.

L-arginine deficiency results in cell cycle arrest via the downregulation of CD3 $\zeta$, a key component of the TCR (*Rodriguez et al., 2002*), and T cell proliferation is impaired in the absence of L-arginine (*Rodriguez et al., 2007*). In line with these observations, we found that improved control of viral replication in mice lacking Arg1+ CD4+ T cells was associated with increased numbers of MCMV-specific effector CD4+ T cells, which are known to limit viral replication in the SGs (*Lucin et al., 1992*; *Walton et al., 2011*). Accordingly, the acquisition of Arg1 expression by highly differentiated CD4+ T cells impinges on antiviral immunity, representing an inhibitory mechanism that operates alongside the production of IL-10 (*Clement et al., 2016*; *Humphreys et al., 2007*). It is also notable here that regulatory T cells may act similarly via the expression of Arg2 (*Lowe et al., 2019*).

IL-10 is regulated by a number of transcription factors, including those required for T cell differentiation, and is induced downstream of the TCR (*Saraiva et al., 2020*). Our discovery that T-bet promotes the development of IL-10-producing CD4+ T cells aligns with the concept that chronic antigen stimulation can trigger a negative feedback loop via these mechanisms, which concurrently drive activation and differentiation. It nonetheless remains less clear to what extent T-bet directly orchestrates the expression of other inhibitory molecules by CD4+ T cells. There was no notable

increase in chromatin-accessible binding motifs for T-bet within either the *Il10* or *Arg1* genes in IL-10-producing CD4$^+$ T cells. However, we did find that deletion of T-bet after acute infection limited the accumulation of highly differentiated CD4$^+$ T cells, including those expressing Arg1 and IL-10. In line with this observation, which suggested a differentiation-linked process of functional remodeling driven by chronic antigen exposure, T-bet is known to promote the differentiation of CD8$^+$ T cells (*Dominguez et al., 2015*; *Omilusik et al., 2015*) and inhibit the expression of TCF1/7 (*Omilusik et al., 2015*) and IL-7R (*Intlekofer et al., 2007*), both of which were downregulated among Arg1$^+$ IL-10-producing CD4$^+$ T cells induced by MCMV.

The relationship between mucosal IL-10-producing CD4$^+$ T cells and classical Tr1 cells requires further clarification. In contrast to Tr1 cells (*Gagliani et al., 2013*), IL-10-producing CD4$^+$ T cells in the SGs do not coexpress CD49b and LAG-3 (*Clement et al., 2016*), highlighting the diverse nature of regulatory CD4$^+$ T cells defined by the secretion of IL-10 (*Brockmann et al., 2018*). However, these cells do express multiple inhibitory receptors, including LAG-3, alongside other markers that typically characterize Tr1 cells, such as CCR5 and granzyme K (*Brockmann et al., 2018*; *Roessner et al., 2021*; *Thelen et al., 2023*). In our model, Thy1.1$^+$ CD4$^+$ T cells also expressed Eomes, akin to clonally expanded T-bet-dependent Tr1 cells (*Thelen et al., 2023*). Accordingly, mucosal IL-10-producing CD4$^+$ T cells may develop along similar lines to conventional Tr1 cells, which differentiate under the control of Eomes in conjunction with Blimp-1 (*Roessner et al., 2021*; *Zhang et al., 2017*).

The mechanisms that directly induce the expression of *Il10* and *Arg1* in CD4$^+$ T cells remain obscure. Although IL-10-producing CD4$^+$ T cells were characterized by prominent clonal expansions, repertoire analysis revealed no obvious determinative role for the TCR. However, the transcription factor c-Maf, which promotes the expression of IL-10 in multiple T$_H$ cell subsets (*Gabryšová et al., 2018*), is known to be upregulated in IL-10-producing CD4$^+$ T cells during chronic infection with MCMV (*Clement et al., 2016*). We also demonstrated previously that ICOS signaling promotes the accumulation of IL-10-producing CD4$^+$ T cells under the same conditions (*Clement et al., 2016*), and c-Maf acts downstream of ICOS (*Bauquet et al., 2009*; *Nurieva et al., 2003*). It therefore seems likely that an ICOS–c-Maf axis participates in the direct induction of *Il10*, although it is less clear how this applies to *Arg1*. Moreover, we found that Arg1$^+$ IL-10-producing CD4$^+$ T cells developed optimally in vitro in response to high-dose antigen stimulation in the presence of IL-12, suggesting determinative roles for ERK MAP kinase and the transcription factor STAT4 (*Saraiva et al., 2009*).

The evolutionary advantage of a mechanism that leads to the production of immune regulatory molecules and consequently facilitates viral persistence without impacting autoimmunity is difficult to explain. One possibility is that some of these molecules, potentially including Arg1, serve to maintain tissue health in the presence of ongoing inflammation. It would be informative in this context to evaluate how Arg1 delivery via CD4$^+$ T cells impacts mucosal pathology over time at sites beyond the SGs. Alternatively, local immune suppression may somehow limit the emergence of more pathogenic viral strains under conditions of persistent replication, thereby protecting the host and the wider population. Irrespective of the precise explanation, it seems clear that viral persistence can be facilitated by this intrinsic regulatory mechanism, a phenomenon that could feasibly extend to pathogens other than MCMV.

In summary, we have demonstrated that T-bet activity during a chronic viral infection can impede antiviral immune control by driving the development of highly differentiated T$_H$1-like cells that express genes encoding inhibitory molecules, including IL-10 and Arg1. Importantly, these Arg1$^+$ IL-10-producing CD4$^+$ T cells developed in vitro under conditions of extreme antigen stimulation, especially in the presence of IL-12, and the expression of Arg1, which was also secreted in vivo as a soluble immune regulatory protein, facilitated viral replication during the chronic phase of infection with MCMV. These observations are conceivably important not only from a biological perspective but also from a translational perspective, revealing a previously unappreciated mechanism through which CD4$^+$ T cells can suppress potentially harmful immune responses via the regulation of L-arginine.

## Materials and methods
### Mice
10BiT reporter mice were originally derived by Padraic Fallon (Trinity College Dublin, Republic of Ireland). These mice express CD90/CD90.1 (Thy1.1) under the control of the *Il10* promoter and retain

endogenous expression of IL-10 (**Maynard et al., 2007**). $Cd4^{+/+}Tbx21^{flox/flox}$ mice and $Cd4^{CreERT2/+}T$-$bx21^{flox/flox}$ mice were generated by crossing $Rosa26^{flSTOPtdRFP}$ mice (**Luche et al., 2007**) with $Tbx21^{flox/flox}$ mice (**Intlekofer et al., 2008**). $Arg1^{flox/flox}$ mice (JAX stock #008817) and $Lyz2^{Cre/+}$ mice (JAX stock #004871) were purchased from The Jackson Laboratory. These mice were bred to generate $Lyz2^{+/+}Arg$-$1^{flox/flox}$ mice and $Lyz2^{Cre/+}Arg1^{flox/flox}$ mice. $Arg1^{flox/flox}$ mice were further bred with $Cd4^{Cre/+}$ mice to generate $Cd4^{+/+}Arg1^{flox/flox}$ mice and $Cd4^{Cre/+}Arg1^{flox/flox}$ mice. C57BL/6 WT mice (JAX stock #000664) were purchased from Charles River or Envigo. OT-II mice (JAX stock #004194) were purchased from The Jackson Laboratory. $Cd4^{Cre/+}$ mice (JAX stock #022071) were a kind gift from Sarah Dimeloe (University of Birmingham, UK).

## Infections and treatments

MCMV was prepared via sorbital gradient purification as described previously (**Stacey et al., 2014**). Mice were infected with $3 \times 10^4$ pfu of MCMV intraperitoneally. $Cd4^{+/+}$ and $Cd4^{Cre/+}$ mice were injected intraperitoneally with tamoxifen (Sigma-Aldrich) as indicated for 5 days from day 7 p.i. at a daily dose of 75 mg/kg (20 mg/ml). In some experiments, mice were injected intraperitoneally with anti-CD4 (100 µg of clone GK1.5 and 100 µg of clone YTS191) and/or anti-CD8 (100 µg of clone YTS169.4 and 100 µg of clone YTS156.7.7) or an isotype control, rat IgG2b anti-keyhole limpet hemocyanin (400 µg of clone LTF-2), on days 6 and 8 p.i. (all from BioXell).

## Next-generation sequencing

Leukocytes were isolated directly ex vivo from the SGs of 10BiT mice on day 14 p.i. (minimum $n = 5$ mice per group with three replicates). Pooled cells were labeled using a Zombie Aqua Fixable Viability Kit (BioLegend) and stained with anti-CD16/CD32 (Fc block, BioLegend) followed by anti-CD4–BV605 (clone RM4-5, BioLegend), anti-CD44–FITC (clone IM7, BioLegend), anti-CD62L–PE-Cy7 (clone MEL-14, BioLegend), and anti-CD90/90.1–PE (clone OX-7, BioLegend). Cells were sorted as CD4+ CD44+ CD62L− CD90/90.1+ (Thy1.1+) or CD90/90.1− (Thy1.1−) populations directly into Buffer RLT or Buffer RLT Plus (QIAGEN) using a modified FACS Aria II (BD Biosciences). Total RNA was extracted using an RNeasy Micro Kit or an RNeasy Mini Kit (QIAGEN), and RNA integrity scores were determined using an RNA 6000 Pico Kit (Agilent).

## RNA-seq

Multiplexed mRNA libraries were obtained using a TruSeq RNA Library Prep Kit v2 (Illumina) and analyzed using a Bioanalyzer High Sensitivity DNA Chip (Agilent). Libraries were sequenced using a HiSeq 2500 System (Illumina). Paired-end reads (100 bp) were trimmed using Trim Galore (https://www.bioinformatics.babraham.ac.uk/projects/trim_galore/) and assessed for quality using FastQC (https://www.bioinformatics.babraham.ac.uk/projects/fastqc/). Reads were mapped to the mouse GRCm38 reference genome using STAR (**Dobin et al., 2013**). Counts were assigned to transcripts using featureCounts (**Liao et al., 2014**) with the GRCm38.84 Ensembl gene build GTF (http://www.ensembl.org/info/data/ftp/index.html/). Differential gene expression analyses were performed using DESeq2 (**Love et al., 2014**). Genes were discarded from the analysis if differential expression failed to reach significance (adjusted $p < 0.05$ with Benjamini–Hochberg correction).

## ATAC-seq

ATAC-seq was performed as described previously (**Buenrostro et al., 2013**) using a Nextera DNA Sample Preparation Kit (Illumina). DNA was isolated after library preparation using a MiniElute PCR Purification Kit (QIAGEN). Size selection was performed using a BluePippin System (Sage Science) with 2% Agarose Gel Cassettes (Sage Science). Libraries were sequenced using a HiSeq 4000 System (Illumina). Paired-end reads (100 bp) were trimmed using Trim Galore (https://www.bioinformatics.babraham.ac.uk/projects/trim_galore/) and assessed for quality using FastQC (https://www.bioinformatics.babraham.ac.uk/projects/fastqc/). Reads were mapped to the mouse GRCm38 reference genome using BWA (**Li and Durbin, 2009**). Duplicate reads were removed using MarkDuplicates (http://broadinstitute.github.io/picard/). Peaks were called using MACS2 (**Zhang et al., 2008**) in BAMPE mode (adjusted $p < 0.05$ with Benjamini–Hochberg correction). Differential analysis of open regions was performed using DiffBind (http://bioconductor.org/packages/release/bioc/vignettes/DiffBind/inst/doc/DiffBind.pdf).

## TCR-seq

Unique molecular identifier (UMI)-labeled 5′-RACE TCR cDNA libraries were synthesized using a Mouse TCR Profiling Kit (MiLaboratories). Indexed samples were pooled and sequenced using a MiSeq System (Illumina). Paired-end reads (150 bp) were extracted and clustered by UMI using MiGEC (*Shugay et al., 2014*). Sequences were discarded from the analysis if the read count was <4 per cDNA. Error correction was performed using MiGEC. Repertoires were extracted using MiXCR (*Bolotin et al., 2015*). The weighted average use of bulky, charged, and strongly interacting (aromatic and hydrophobic) amino acids positioned centrally in the CDR3α/β sequences and *Trbv* gene use (weighted by frequency) were calculated using VDJtools (*Shugay et al., 2015*). Diversity was calculated by downsampling the repertoires to an equal number of UMIs ($n = 4300$ for TCRα and $n = 3900$ for TCRβ) three separate times and plotting the mean Chao1 index, reflecting lower bound total diversity, and (1 − normalized Shannon–Wiener index), reflecting clonality. Visualization was achieved using PlotQuantileStats in VDJtools. The mean number of unique nucleotide sequences encoding the same amino acid sequence (convergence) was calculated for the 2000 most prevalent clonotypes in each sample using VDJtools. Overlap was calculated for the 2000 most prevalent clonotypes using F2 metrics to estimate sharing at the amino acid level among *V* gene-matched sequences in each sample. Cluster analysis was performed using the 500 most prevalent clonotypes in pooled samples (Thy1.1+ versus Thy1.1−). These datasets were analyzed using tcrgrapher (https://github.com/KseniaMIPT/tcrgrapher), an R library based on ALICE (*Pogorelyy et al., 2019*). The real and expected numbers of neighbors were calculated for each clonotype with a maximum of $n = 1$ amino acid mismatch. Clonotypes with a significantly higher number of neighbors than expected on statistical grounds (adjusted $p < 0.0001$ with Benjamini–Hochberg correction) were identified as tcrgrapher hits. The expected number of neighbors was estimated via generation probabilities calculated for each clonotype using OLGA (*Sethna et al., 2019*), with the selection factor set at $Q = 6.27$ (*Elhanati et al., 2018*).

## Bioinformatics

RNA-seq analysis was performed using DESeq2/1.32.0, dplyr/1.0.7, and GenomicRanges/1.44.0. Volcano plots were drawn using ggplot2/3.3.5 and ggrepel/0.9.1. TCR-seq analysis was performed using MiGEC/1.2.9, MiXCR/3.0.13, VDJtools/1.2.1, tidyverse/1.3.1, igraph/1.2.6, ggnetwork/0.5.10, msa/1.22.0, tcrgrapher/0.0.09000, stringdist/0.9.6.3, and ggseqlogo/0.1. Other software packages used across next-generation sequencing platforms included Nextflow (https://www.nextflow.io/) (*Di Tommaso et al., 2017*), trimgalore/0.6.4 (https://www.bioinformatics.babraham.ac.uk/projects/trim_galore/), FastQC/0.11.8 (https://www.bioinformatics.babraham.ac.uk/projects/fastqc/), multiqc/1.7 (https://multiqc.info/) (*Ewels et al., 2016*), STAR/2.7.3 (https://github.com/alexdobin/STAR) (*Dobin et al., 2013*), BWA/0.7.10 (http://bio-bwa.sourceforge.net/) (*Li and Durbin, 2009*), picard/2.20.2 (http://broadinstitute.github.io/picard/), samtools/1.9 (http://www.htslib.org/) (*Danecek et al., 2021*), bamtools/2.5.1 (https://github.com/pezmaster31/bamtools) (*Barnett et al., 2011*), featurecounts/2.0.0 (http://subread.sourceforge.net/) (*Liao et al., 2014*), and MACS2/2.1.2 (https://pypi.org/project/MACS2/) (*Zhang et al., 2008*). Venn diagrams were drawn using InteractiVenn (http://www.interactivenn.net) (*Heberle et al., 2015*). Heatmaps were drawn using Morpheus (https://software.broadinstitute.org/morpheus). ATACseq data were visualized using Integrative Genomics Viewer (*Robinson et al., 2011*). Gene ontology analysis was performed using GOTermFinder (https://go.princeton.edu/cgi-bin/GOTermFinder). ATAC-seq motif analysis was performed using The MEME Suite (https://meme-suite.org/meme/doc/cite.html?man_type=web) (*Bailey et al., 2015*). *Tbx21* binding motifs were obtained using JASPAR[2020] (*Fornes et al., 2020*).

## Quantification of MCMV

Infectious virus was quantified in organs via plaque assay as described previously (*Stack et al., 2015*). 3T3 cells were purchased directly from the American Type Culture Collection. *Mycoplasma* infection was excluded prior to assay. Viral loads in oral lavage were quantified via qPCR (*Clement et al., 2016*; *Kamimura and Lanier, 2014*).

## Western blotting

Leukocytes were isolated from SGs and spleens as described previously (*Stacey et al., 2011*). CD4+ T cells were purified via magnetic separation using a MagniSort Mouse CD4 Positive Selection Kit

(Thermo Fisher Scientific). Cell lysates were generated from equal numbers of cells using NuPAGE LDS Sample Buffer (Thermo Fisher Scientific) supplemented with 100 mM dithiothreitol (Sigma-Aldrich). Samples were loaded onto 4–12% NuPAGE Bis-Tris Gels (Thermo Fisher Scientific) after boiling and transferred to a PVDF membrane using an XCell II Blot Module (Thermo Fisher Scientific). Blots were probed with anti-arginase-1 (rabbit polyclonal, Thermo Fisher Scientific) and developed using anti-rabbit IgG–HRP (Bio-Rad). Band intensity was determined using a G:BOX Gel Imaging System (Syngene). Blots were then stripped using Restore PLUS Western Blot Stripping Buffer (Thermo Fisher Scientific) and probed again with anti-actin (rabbit polyclonal, Abcam).

## Flow cytometry

Leukocytes were isolated from SGs and spleens as described previously (*Stacey et al., 2011*). Cells were labeled using a Zombie Aqua Fixable Viability Kit (BioLegend) and stained with anti-CD16/CD32 (Fc block, BioLegend) followed by combinations of anti-CD4–APC-Cy7, anti-CD4–BV605, or anti-CD4–Pacific Blue (clone RM4-5, BioLegend), anti-CD8–BV605 (clone 53.6.7, BioLegend), anti-CD39–Alexa Fluor 647 (clone Duha59, BioLegend), anti-CD44–FITC (clone IM7, BioLegend), anti-CD62L–BV711 or anti-CD62L–PE-Cy7 (clone MEL-14, BioLegend), anti-CD90/90.1–APC, anti-CD90/90.1–BV605, or anti-CD90/90.1–PE (clone OX-7, BioLegend), anti-CD127–BV711 (clone A7R34, BioLegend), anti-CD223–PE (LAG-3) (clone C9B7W, BioLegend), anti-NK1.1–PE (clone PK136, BioLegend), anti-PD-1–BV421 (clone 29F.1A12, BioLegend), anti-TIGIT–APC (clone 1G9, BioLegend), and anti-TIM-3–BV711 or anti-TIM-3–PerCP-Cy5.5 (clone RMT3-23, BioLegend). Tetramer staining was performed as described previously (*Clement et al., 2016*). The following specificities were used in this study: H-2D$^b$ M45 residues 985–993 (HGIRNASFI), H-2K$^b$ IE3 residues 416–423 (RALEYKNL), H-2K$^b$ M38 residues 316–323 (SSPPMFRV), and H-2K$^b$ m139 residues 419–426 (TVYGFCLL) (National Institutes of Health Tetramer Core Facility). Internal antigen expression was determined after fixation/permeabilization in BD Cytofix/Cytoperm Solution (BD Biosciences) or FoxP3/Transcription Factor Staining Buffer (eBioscience). Cells were stained with combinations of anti-arginase-1–APC (clone Met1-Lys322, Bio-Techne) or anti-arginase-1–PE-Cy7 (clone A1ex5, eBioscience), anti-Ki-67–Alexa Fluor 488 (clone 11F6, BioLegend), anti-T-bet–BV605 or anti-T-bet–Pacific Blue (clone 4B10, BioLegend), and anti-TCF1/7–Alexa Fluor 647 (clone C63D9, Cell Signaling Technology). Cells were acquired using an Attune NxT (Thermo Fisher Scientific) or an LSR Fortessa (BD Biosciences). All flow cytometry data were analyzed using FlowJo v10.5.3 or v10.8.1 (FlowJo LLC).

## Quantification of anti-SSA

Plasma samples were obtained from cardiac punctures and assessed for anti-SSA/Ro 60 IgG levels via ELISA (Alpha Diagnostics).

## In vitro induction of Arg1

Spleens were harvested from OT-II mice (*n* = 13) and digested for 30 min via the injection of collagenase-D (Thermo Fisher Scientific). Cells were isolated and purified via magnetic separation using a Pan Dendritic Cell Isolation Kit (Miltenyi Biotec) and a MagniSort Mouse CD4 Positive Selection Kit (Thermo Fisher Scientific). Cultures were established using $2 \times 10^4$ purified CD11c$^+$ dendritic cells/ml and $1 \times 10^5$ purified CD4$^+$ T cells/ml as described previously (*Saraiva et al., 2009*). Cells were cultured in the absence or presence of IL-12 (3 ng/ml, PeproTech) without or with the OVA peptide ISQAVHAAHAEINEAGR (residues 323–339, GenScript) at a concentration of 0.01 or 0.5 µM in RPMI medium supplemented with 100 U/ml penicillin, 100 µg/ml streptomycin, 2 mM L-glutamine, and 10% heat-inactivated fetal calf serum (all from Thermo Fisher Scientific). On day 7, cells were unexposed (medium control) or exposed to plate-bound anti-CD3 (2 µg/ml, clone 145-2C11, BioLegend) and anti-CD28 (2 µg/ml, clone 37.51, BioLegend). After 2 hr, the cultures were supplemented without or with brefeldin A (2 µg/ml, Sigma-Aldrich), and after a further 2 hr, cells were labeled using a Zombie Aqua Fixable Viability Kit (BioLegend) and stained with anti-CD16/CD32 (Fc block, BioLegend) followed by anti-CD4–Pacific Blue (clone RM4-5, BioLegend) and anti-CD8–BV605 (clone 53.6.7, BioLegend). Internal antigen expression was determined after fixation/permeabilization in BD Cytofix/Cytoperm Solution (BD Biosciences). Cells were stained with anti-arginase-1–PE-Cy7 (clone A1ex5, eBioscience), anti-IFNγ–FITC (clone XMG1.2, BioLegend), and anti-IL-10–APC (clone JES5-16E3, eBioscience). Cells were acquired using an Attune NxT (Thermo Fisher Scientific), and data were analyzed using FlowJo

v10.5.3 (FlowJo LLC). Supernatants were harvested from cultures that lacked brefeldin A after a further 48 hr and analyzed for Arg1 protein expression via ELISA (Aviva Systems Biology).

## Functional assays

Leukocytes from SGs and spleens were stimulated with peptides at a final concentration of 3 µg/ml for 2 hr at 37°C. The following peptides were used to stimulate CD4$^+$ T cells: m09 residues 133–147 (GYLYIYPSAGNSFDL), M25 residues 409–423 (NHLYETPISATAMVI), m139 residues 560–574 (TRPYRYPRVCDASLS), and m142 residues 24–38 (RSRYLTAAAVTAVLQ). The cultures were then supplemented with brefeldin A (2 µg/ml, Sigma-Aldrich) and incubated for a further 4 hr at 37°C. After stimulation, cells were labeled using a Zombie Aqua Fixable Viability Kit (BioLegend) and stained with anti-CD16/CD32 (Fc block, BioLegend) followed by anti-CD4–APC-Cy7 or anti-CD4–BV605 (clone RM4-5, BioLegend). Internal antigen expression was determined after fixation/permeabilization in BD Cytofix/Cytoperm Solution (BD Biosciences). Cells were stained with combinations of anti-IFNγ–APC-Cy7, anti-IFNγ–FITC, or anti-IFNγ–Pacific Blue (clone XMG1.2, BioLegend), anti-IL-10–APC or anti-IL-10–FITC (clone JES5-16E3, eBioscience), and anti-T-bet–BV605 (clone 4B10, BioLegend). Cells were acquired using an Attune NxT (Thermo Fisher Scientific), and data were analyzed using FlowJo v10.5.3 (FlowJo LLC).

## Statistics

Sample sizes for next-generation sequencing experiments were calculated using G*Power (https://www.psychologie.hhu.de/arbeitsgruppen/allgemeine-psychologie-und-arbeitspsychologie/gpower.html), where a minimum of $n = 5$ pooled mice per group was used to detect a difference in means with 90% power and an $\alpha$ value set at 0.05 across a minimum of three replicates. All outliers were included in the final datasets. Comparisons between two groups were performed using the Mann–Whitney $U$ test. Significance across all tests is reported as $*p < 0.05$, $**p < 0.01$, $***p < 0.001$, and $****p < 0.0001$.

## Acknowledgements

We thank Steven Reiner (Columbia University, New York, USA) for providing $Tbx21$$^{flox/flox}$ mice, Joerg Fehling (Ulm University, Germany) for providing $Rosa26$$^{flSTOPtdRFP}$ mice, and Sarah Dimeloe (University of Birmingham, UK) for providing $Cd4$$^{Cre/+}$ mice. Biotinylated monomers were obtained from the NIH Tetramer Core Facility. MC was supported by a Systems Immunity Research Institute Fellowship (Cardiff University). VVK received infrastructure support from the Deutsche Forschungsgemeinschaft (DFG) Cluster of Excellence "Precision Medicine in Chronic Inflammation" (PMI). OVB, DMC, and DAP were supported by a Ministry of Science and Higher Education of the Russian Federation Subsidy Grant (075-15-2019-1789). DRW was supported by a Wellcome Trust Senior Research Fellowship (110199/Z/15/Z). DAP was supported by a Wellcome Trust Senior Investigator Award (100326/Z/12/Z). IRH was supported by a Medical Research Council Confidence in Concept Award, a Wellcome Trust Collaborator Award (209213/Z/17/Z), and a Wellcome Trust Senior Research Fellowship (207503/Z/17/Z).

## Additional information

### Funding

| Funder | Grant reference number | Author |
| --- | --- | --- |
| Wellcome Trust | 207503/Z/17/Z | Ian R Humphreys |
| Wellcome Trust | 100326/Z/12/Z | David A Price |
| Wellcome Trust | 110199/Z/15/Z | David R Withers |
| Deutsche Forschungsgemeinschaft | Cluster of Excellence "Precision Medicine in Chronic Inflammation" | Valeriia V Kriukova |
| Ministry of Science and Higher Education of the Russian Federation | 075-15-2019-1789 | Dmitriy M Chudakov |

| Funder | Grant reference number | Author |
|--------|------------------------|--------|
| Wellcome Trust | 209213/Z/17/Z | Ian R Humphreys |

The funders had no role in study design, data collection, and interpretation, or the decision to submit the work for publication. For the purpose of Open Access, the authors have applied a CC BY public copyright license to any Author Accepted Manuscript version arising from this submission.

## Author contributions

Mathew Clement, Conceptualization, Data curation, Formal analysis, Validation, Investigation, Methodology, Writing – original draft, Project administration, Writing – review and editing; Kristin Ladell, Morgan Marsden, Anna Cardus Figueras, Valeriia V Kriukova, Ksenia R Lupyr, Data curation, Formal analysis; Kelly L Miners, Lucy Chapman, Jake Scott, Simon Clare, Data curation; Robert Andrews, Simon A Jones, Resources, Formal analysis; Olga V Britanova, Dmitriy M Chudakov, Data curation, Formal analysis, Funding acquisition, Writing – review and editing; David R Withers, Resources, Formal analysis, Funding acquisition; David A Price, Resources, Formal analysis, Supervision, Funding acquisition, Methodology, Writing – original draft, Writing – review and editing; Ian R Humphreys, Conceptualization, Resources, Formal analysis, Supervision, Funding acquisition, Methodology, Writing – original draft, Project administration, Writing – review and editing

## Author ORCIDs

Mathew Clement ⬛ http://orcid.org/0000-0002-9280-5281
David A Price ⬛ http://orcid.org/0000-0001-9416-2737
Ian R Humphreys ⬛ http://orcid.org/0000-0002-9512-5337

## Ethics

All mouse experiments were approved by the Biological Services Facility (Cardiff University) and performed under UK Home Office Project License P7867DADD.

## Decision letter and Author response

Decision letter https://doi.org/10.7554/eLife.79165.sa1
Author response https://doi.org/10.7554/eLife.79165.sa2

# Additional files

## Supplementary files

- MDAR checklist

## Data availability

All data needed to evaluate the conclusions of this study are presented in the paper and/or the Supplementary Materials. RNA-seq data generated as part of this study have been deposited in ArrayExpress (E-ERAD-445). A fully annotated version of the RNA-seq dataset is freely available via Zenodo (https://doi.org/10.5281/zenodo.7243956). An annotated list of genes shared between the dataset presented here (E-ERAD-445) and the in vitro $T_H1$ dataset (E-MTAB-2582) is also freely available via Zenodo (https://doi.org/10.5281/zenodo.7447477). TCR-seq data generated as part of this study have been deposited in the NCBI Sequence Read Archive (PRJNA860054). Processed repertoire datasets are freely available via Figshare (https://figshare.com/projects/Clement_InhibitoryCD4/143541).

The following datasets were generated:

| Author(s) | Year | Dataset title | Dataset URL | Database and Identifier |
|-----------|------|---------------|-------------|-------------------------|
| Clement M, Humphreys IR | 2021 | Characterization of IL-10-producing CD4+ T cells in a mucosal herpesvirus infection | https://www.ebi.ac.uk/arrayexpress/experiments/E-ERAD-445/ | ArrayExpress, E-ERAD-445 |

*Continued on next page*

*Continued*

| Author(s) | Year | Dataset title | Dataset URL | Database and Identifier |
|---|---|---|---|---|
| Clement M | 2023 | TCR-seq data generated as part of this study | https://www.ncbi.nlm.nih.gov/sra/?term=PRJNA860054 | NCBI Sequence Read Archive, PRJNA860054 |
| Clement M | 2022 | RNA-seq data comparing salivary gland IL-10-producing CD4+ T cells versus IL-10-nonproducing CD4+ T cells on day 14 after infection with MCMV | https://doi.org/10.5281/zenodo.7243956 | Zenodo, 10.5281/zenodo.7243956 |
| Clement M | 2022 | Shared gene use between salivary gland IL-10-producing CD4+ T cells and T$_H$1-like CD4+ T cells generated in vivo | https://doi.org/10.5281/zenodo.7447477 | Zenodo, 10.5281/zenodo.7447477 |
| Kriukova V | 2022 | metadata_Clement_TCR | https://doi.org/10.6084/m9.figshare.20310990.v1 | figshare, 10.6084/m9.figshare.20310990.v1 |

The following previously published datasets were used:

| Author(s) | Year | Dataset title | Dataset URL | Database and Identifier |
|---|---|---|---|---|
| Stubbington M, Mahata B, Svensson V, Deonarine A, Nissen JK | 2015 | An mRNA-sequencing atlas of mouse CD4+ T cell transcriptomes | https://www.ebi.ac.uk/arrayexpress/experiments/E-MTAB-2582/ | ArrayExpress, E-MTAB-2582 |
| Zhu J, Jankovic D, Oler AJ, Wei G, Sharma S, Hu G, Guo L, Yagi R, Yamane H, Punkosdy G, Feigenbaum L, Zhao K, Paul WE | 2012 | The transcription factor T-bet is induced by multiple pathways and prevents an endogenous Th2 cell program during Th1 cell responses | https://www.ncbi.nlm.nih.gov/geo/query/acc.cgi?acc=GSE38808 | NCBI Gene Expression Omnibus, GSE38808 |

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
