## [Editor Report]

This study analyzes the development and functional relevance of IL-10-producing regulatory T cells in a mouse model of cytomegalovirus infection. The results indicate that IL-10-producing CD4^+^ T cells express genes associated with chronically activated T_H_1-like cells, undergo clonal expansion, and inhibit antiviral T cell responses via the secretion of arginase, an enzyme that breaks down an amino acid required for T cell activation and proliferation. These findings reveal a novel and important immunoregulatory mechanism that facilitates viral persistence.

---

## [Decision Letter]

**Decision letter after peer review:**

Thank you for submitting your article "Inhibitory IL-10-producing CD4^+^ T cells are T-bet-dependent and facilitate cytomegalovirus persistence via coexpression of arginase-1" for consideration by *eLife*. Your article has been reviewed by 3 peer reviewers, one of whom is a member of our Board of Reviewing Editors, and the evaluation has been overseen by Satyajit Rath as the Senior Editor. The reviewers have opted to remain anonymous.

The reviewers have discussed their reviews with one another, and since they raise distinct crucial questions, it is requested that you respond, either with additional evidence and/or with clarifications, to the individual reviews below.

*Reviewer #1 (Recommendations for the authors):*

Although the results of the present study are interesting and potentially relevant, some questions remained unanswered.

1. Although it is well known that CD4 T cells, rather than CD8 cells, are pivotal in controlling MCMV in the SG, the use of Cd4cre mice might have also affected the functionality of CD8 cells. This question should be experimentally addressed.

2. It remains unclear how Arg-1 expression was regulated and whether this effect is specific/limited to the salivary gland. What is the experimental evidence that frequent antigen exposure triggers Arg1 expression?

3. Does Arg1 causes MCMV persistence, or does the virus persistence in the SG trigger the formation of IL-10-producing Arg1 positive CD4 T cells? Since MCMV can persist for months in the salivary glands, even in immunocompetent mice, the dynamics of IL-10-producing CD4 T cells during and after the resolution of persistent infection should be investigated.

4. It is well established that a single passage of tissue-culture-derived MCMV is sufficient to result in a highly virulent MCMV. Therefore, comparing the virulence of salivary gland-derived MCMV from Arg1 positive and Arg1 negative mice would be informative.

*Reviewer #2 (Recommendations for the authors):*

The novelty of the findings are limited by a recent paper (JCI Insight. 2019 Dec 19; 4(24): e129756) that shows that Arginase-2 in FOXP3+Tregs contributes to suppression. This study has to be quoted and discussed.

Il-10-T-cells in Figure 1 and 2 might contain bystander cells that could even be entirely responsible for the differences in gene expression and clonal expansions. This needs to be ruled out. IFN-g producing Il-10- cells would have been much more convincing. A list of differentially expressed genes is missing, how were the shown genes selected, by p-value, fold expression or arbitrarily?

Figure 3: Inducible CD4 Cre mice would have been much more convincing to rule out a contribution of CD8^+^T-cells (it is stated in the Discussion that CD8^+^T-cells do not express arginase, but this data needs to be shown). It must also be controlled if arginase is expressed in IL-10- cells, otherwise the Figure title is simply an overstatement. CD4-Cre mice without the floxed alleles need to be analysed as controls.

Figure 4//5: Based on the literature a similar role could be expected for the related T-box transcription factor Eomesodermin, the choice of T-bet seems a bit arbitrary. T-bet ko cells fail to differentiate to effector cells, the molecular mechanism of arginase-1 induction remains therefore unclear, as also admitted by the authors.

*Reviewer #3 (Recommendations for the authors):*

Figure 1:

To which extent Thy1.1 expression correlates with IL-10 protein staining in the SG of MCMV-infected mice? This should be provided in the supplement. Are all of these, or the majority of these cells, antigen-specific, i.e. what would be the correlation with for example available M25 tetramer staining or peptide pool restimulation?

It remains unclear to which extent the phenotype of in vivo generated IL-10+CD4^+^T cells overlaps with conventional in vivo generated Th1 cells (maybe even ones of acutely MCMV-infected mice), cytotoxic Th1 cells (for example lung cytotoxic Th1 cells) and Tr1-like cells (either generated in vitro with IL-27 and TGFb or isolated from an in vivo setting). Which gene families constitute 1646 DEG in Figure 1E.

The authors could also co-stain IFNγ and GzmB with Thy1.1 after polyclonal or even antigen-stimulation in vitro.

Some genes associated with the induction of IFN-g, including Il18r1 and il18rap, were downregulated in Thy1.1+ CD4^+^ T cells, however IFNγ does not seem to be differentially regulated, or? While Thy1.1+ cells express a module of inhibitory genes, their expression of Tcf1/Tcf7 seems to be downregulated? Is any difference in the expression of Tox observed?

Figure 2:

The authors show that the Thy1.1+ pool shows a narrower and more convergent repertoire, which could be an indication of a clonally expanded population. VDJtools, which the authors use, offers a more comprehensive diversity estimation with PlotQuantileStats. This would enable the authors to visualize the repertoire clonality of Thy1.1+ vs Thy1.1- pools.

For better understanding, is the clonality analysis performed on a sample derived of several pooled mice? Have the authors had the opportunity to assess the degree of clone sharing between e.g., splenic and salivary gland cells?

Figure 3:

The authors demonstrate higher transcript levels and open chromatin structures of Arg1 in Thy1.1+ cells. They should provide a co-staining of Arg-1 and Thy1.1 on a protein level.

Although CD4 T cells are critical in controlling MCMV in the SG, Cd4cre mice target also CD8 and NKT cells can the authors confirm there is no Arginase-1 expression in these cells as well that could contribute to MCMV control in SG? This could potentially be solved by adoptive transfer of WT vs. Arg1-/- CD4 T cells. Otherwise, conclusion of this figure should be downplayed as it is not clear that Arg1 expression by CD4^+^ t cells selectively inhibit accumulation of virus-specific CD4^+^ and CD8^+^ T cells.

Is there any impact on the IL-10 expression by CD4^+^ T cells observed in the SG of CD4Cre Arg1flox mice infected with MCMV?

NK cells are known to contribute to the viral control in SG and TRAIL+ NK cells capable of killing activated CD4^+^ T cells accumulate in the SG during CMV infection (Schuster et al. 2011). Do the authors observe any differences in the expression of DR5 in relation to Thy1.1?

Are there any functional and phenotypical advantages in the CD4^+^ T cell compartment of uninfected mice?

The proliferation effect observed in CD4CreArg1flox mice – can it be reversed by Arginase-1 supplementation?

Figure 4:

In figure 3A, are the CD4 T cells pre-gated on CD44? One would assume that the majority of CD4^+^ T cells in the SG during CMV infection would be T-bet+, i.e., of Th1 phenotype. Can this be attributed to the technical issue of T-bet staining? Also does the FMO/Ab staining come from the same mouse as the % of Thy1.1 cells seem to increase from upper (11.9%) to lower (19.43%) plot?

Figure 5:

In this figure the authors demonstrate that IFNγ-expressing cells are increased with T-bet deficiency and as a result the virus is cleared better. What is the correlation between IFN-g and Arg-1 expression? Are the virus specific IFNγ-producing cells in T-bet-deficient mice also of CD44loPD-1loCD62Lhi phenotype? The title of the figure says that IL-10-producing CD4^+^ T cells develop in a T-bet-dependent manner, however the authors do not probe for IL-10 expression here.

What formal proof demonstrates that repetitive antigen exposure results in Arginase-1 upregulation? This could potentially be mimicked in vitro (Saraiva et al., 2009).

In Figure 5H, significance in viral clearance seems to be driven by a number of plaques observed in a sample of one mouse. Even like that, the difference does not reach even half a log a difference. I suggest increasing n to strengthen this result. In contrast to Cd4Cre Tbet-flox mice which seem to compensate for the lack of T-bet by day 28 there is no protective effect, viral control is still observed in Cd4Cre Arg1flox (this paper) and CD4Cre IL10-flox (author´s previous publication Clement et al., 2016). How do the authors comment on this?

[Editors' note: further revisions were suggested prior to acceptance, as described below.]

Thank you for resubmitting your work entitled "Inhibitory IL-10-producing CD4^+^ T cells are T-bet-dependent and facilitate cytomegalovirus persistence via coexpression of arginase-1" for further consideration by *eLife*. Your revised article has been evaluated by 2 reviewers, Satyajit Rath (Senior Editor) and a Reviewing Editor.

Although the conclusion was that the manuscript has been improved, there are some remaining issues that need to be addressed, as outlined below. Please note that we do not expect you to add new data, but to discuss your findings in the light of suggestions which can be found below.

*Reviewer #2 (Recommendations for the authors):*

In the revision, the authors have performed all requested experiments.

I do however not think that the interpretation of their data is acceptable in its current form, because they ignore a large body of published evidence on IL-10 producing T-cells that is highly relevant to their study. There is already a lot of confusion in the field, and ignoring seminal work performed by others does not contribute to reaching a general consensus on the molecular identity of these cells.

The authors show in supplementary Figure 5 that IL-10 producing regulatory T-cells express high levels of Eomesodermin, a rather surprising finding for CD4^+^T-cells, but comment on this data only briefly on p11: "No such difference in expression frequency were observed for the related transcription factor Eomesodermin (Eomes)". This is really a misleading statement since the authors do show that the MFI of Eomes is higher in IL-10 producing T-cells. However, no statistical analysis is provided, and a control with CD4^+^T-cells in the absence of CMV infection (presumably Eomes-negative) is also missing. In any case, the large majority of IL-10+ T-cells express high levels of Eomesodermin, while the staining of T-bet suggests that T-bet expression may be actually lower (of course this may be due to technical issues). Thus, the choice to focus on T-bet rather than on Eomes is in my opinion an arbitrary one. I understand that it would take too much work to perform all relevant experiments both with T-bet and Eomes-deficient T-cells in a revision, but ignoring all the published evidence on Eomesodermin-expressing, IL-10 producing CD4^+^T-cells is simply unacceptable. The authors state moreover in the Introduction that IL-10 producing T-cells in their system "are phenotypically different from type 1 regulatory T-cells (Clement 2016)" In their cited work I could not find a detailed phenotypic characterization, so it is unclear to me how the authors came to this conclusion. Moreover, in the revised article, they document in contrast that the IL-10 producing "inhibitory" T-cells they study do express all markers that are well-known to be associated with Tr1-cells, namely LAG3, PD1 and other checkpoint receptors, CCR5, GzmK etc etc.(Brockmann, Soukou et al. 2018, Roessner, Llao Cid et al. 2021, Thelen, Schipperges et al. 2023) Thus, the relationship of the IL-10 producing "inhibitory" T-cells to Tr1-cells, in particular those that express Eomes, needs to be discussed in detail and the relevant articles have to be cited. Notably, also Eomes-expressing Tr1-cells are T-bet dependent (Zhang, Lee et al. 2017) and clonally expanded (Bonnal, Rossetti et al. 2021).

Bonnal, R. J. P., G. Rossetti, E. Lugli, M, et. al. (2021). "Clonally expanded EOMES(+) Tr1-like cells in primary and metastatic tumors are associated with disease progression." Nat Immunol 22(6): 735-745.

Brockmann, L., S. Soukou, B. Steglich, et. al. (2018). "Molecular and functional heterogeneity of IL-10-producing CD4(+) T cells." Nat Commun 9(1): 5457.

Roessner, P. M., L. Llao Cid, E. Lupar, T, et. al. (2021). "EOMES and IL-10 regulate antitumor activity of T regulatory type 1 CD4(+) T cells in chronic lymphocytic leukemia." Leukemia 35(8): 2311-2324.

Thelen, B., V. Schipperges, P. Knorlein, J. F, et. al. (2023). "Eomes is sufficient to regulate IL-10 expression and cytotoxic effector molecules in murine CD4(+) T cells." Front Immunol 14: 1058267.

Zhang, P., J. S. Lee, K. H. Gartlan, I. S, et. al. (2017). "Eomesodermin promotes the development of type 1 regulatory T (TR1) cells." Sci Immunol 2(10).

---

## [Author Response]

Reviewer #1 (Recommendations for the authors):Although the results of the present study are interesting and potentially relevant, some questions remained unanswered.1. Although it is well known that CD4 T cells, rather than CD8 cells, are pivotal in controlling MCMV in the SG, the use of Cd4cre mice might have also affected the functionality of CD8 cells. This question should be experimentally addressed.

We now show that CD4 depletion but not CD8 depletion impacts Arg1 protein concentrations in salivary gland homogenates during chronic infection with MCMV (Figure 3—figure supplement 3B).

2. It remains unclear how Arg-1 expression was regulated and whether this effect is specific/limited to the salivary gland. What is the experimental evidence that frequent antigen exposure triggers Arg1 expression?

To address this issue, we stimulated OVA-specific transgenic CD4^+^ T cells with high-dose or low-dose antigen ± IL-12, which has been shown previously to induce the expression of IL-10 among Th1 cells, as highlighted by Reviewer 3. Only the combination of high-dose antigen ± IL-12 induced Arg1^+^ IL-10-producing cells (Figure 3D) and the secretion of substantial amounts of Arg1 (Figure 3E). These data suggest that high-dose antigen exposure acts similarly during chronic infection with MCMV. See also pages 9, 10, 15, and 23.

3. Does Arg1 causes MCMV persistence, or does the virus persistence in the SG trigger the formation of IL-10-producing Arg1 positive CD4 T cells? Since MCMV can persist for months in the salivary glands, even in immunocompetent mice, the dynamics of IL-10-producing CD4 T cells during and after the resolution of persistent infection should be investigated.

We showed previously that virus-specific IL10-producing CD4^+^ T cells developed in the salivary glands, peaked on day 14 p.i., and then contracted after infection with MCMV, suggesting a determinative role for initially high levels of viral replication (Figure 1B in Clement *et al.*, PLoS Pathog, 2016). These findings are supported by our new data showing that Arg1^+^ IL-10-producing CD4^+^ T cells develop in vitro in response to high-dose antigen stimulation in the presence of IL-12 (Figure 3E).

4. It is well established that a single passage of tissue-culture-derived MCMV is sufficient to result in a highly virulent MCMV. Therefore, comparing the virulence of salivary gland-derived MCMV from Arg1 positive and Arg1 negative mice would be informative.

We attempted this experiment with the aim of generating virus from *CD4*^cre−^*Arg1*^flox+/+^ and *CD4*^cre+^*Arg1*^flox+/+^ mice. However, we obtained very low titres (<3 × 10^3^ PFU) from pools of 3–5 mice, which were insufficient for challenge experiments to assess in vivo pathogenicity. We believe that >30–50 mice per group would therefore be required to address this possibility, which we feel is perhaps not justified here given that such experiments would not alter the main conclusions of our study. We have nonetheless mentioned this interesting possibility on page 15.

Reviewer #2 (Recommendations for the authors):The novelty of the findings are limited by a recent paper (JCI Insight. 2019 Dec 19; 4(24): e129756) that shows that Arginase-2 in FOXP3+Tregs contributes to suppression. This study has to be quoted and discussed.

We thank the reviewer for drawing our attention to this relevant study, which is now highlighted in context and cited on page 14.

Il-10-T-cells in Figure 1 and 2 might contain bystander cells that could even be entirely responsible for the differences in gene expression and clonal expansions. This needs to be ruled out. IFN-g producing Il-10- cells would have been much more convincing.

To address this possibility, we first identified IFN-γ^+^ CD4^+^ T cells and then compared the expression of two proteins that were preferentially upregulated among Thy1.1^+^ CD4^+^ T cells, namely LAG-3 and PD-1. These proteins were significantly upregulated among IFN-γ^+^ Thy1.1^+^ CD4^+^ T cells versus IFN-γ^+^ Thy1.1^−^ CD4^+^ T cells, arguing against a major role for bystander activation (Figure 1—figure supplement 1E).

A list of differentially expressed genes is missing, how were the shown genes selected, by p-value, fold expression or arbitrarily?

All genes shown are significant at p < 0.05 and were selected for heatmap representation based on relevant pathways identified using gene ontology functions as tabulated and now clarified in the legend for Figure 1D. Our full dataset is available at https://doi.org/10.5281/zenodo.7243956.

Figure 3: Inducible CD4 Cre mice would have been much more convincing to rule out a contribution of CD8^+^T-cells (it is stated in the Discussion that CD8^+^T-cells do not express arginase, but this data needs to be shown).

We did not refer to Arg1 expression by CD8^+^ T cells in the original Discussion. However, we now show that CD8^+^ T cells do indeed express Arg1 (Figure 3—figure supplement 3A), but that only CD4 depletion impacts Arg1 protein concentrations in salivary gland homogenates during chronic infection with MCMV (Figure 3—figure supplement 3B). We also show that CD4^+^ T cells secrete Arg1 in vitro (Figure 3D). These data suggest that CD4^+^ T cells rather than CD8^+^ T cells are the primary source of Arg1 during the chronic phase of infection with MCMV. We discuss these data on page 13.

It must also be controlled if arginase is expressed in IL-10- cells, otherwise the Figure title is simply an overstatement.

The intracellular detection of Arg1 via flow cytometry proved to be suboptimal in our analyses of C57BL/6 mice (discussed on page 13). However, we now show that only conditions that induce IL-10-producing CD4^+^ T cells induce the secretion of substantial amounts of Arg1 (Figure 3E). These data are consistent with the conclusion that the upregulation of Arg1 is a hallmark of inhibitory CD4^+^ T cells, which are also likely the predominant source of secreted Arg1 within the CD4^+^ T cell compartment. We discuss these data on pages 9, 10, and 15. We have also amended the title of the relevant legend for clarity (now Figure 4).

CD4-Cre mice without the floxed alleles need to be analysed as controls.

We now show that viral chronicity is not influenced by the floxed alleles in a comparison of *CD4*^cre+^*Arg*^flox+^ mice and unfloxed CD4^cre+^ mice (Figure 4—figure supplement 4A).

Figure 4//5: Based on the literature a similar role could be expected for the related T-box transcription factor Eomesodermin, the choice of T-bet seems a bit arbitrary.

In contrast to T-bet, we did not observe enriched expression of Eomes protein (Figure 5—figure supplement 5 A, B), as now highlighted on page 11.

T-bet ko cells fail to differentiate to effector cells, the molecular mechanism of arginase-1 induction remains therefore unclear, as also admitted by the authors.

We now include new data showing that high-dose antigen drives the development of Arg1^+^ CD4^+^ T cells and Arg1^+^ IL-10-producing CD4^+^ T cells (Figure 3D, E). The putative molecular mechanisms are now discussed more fully on page 16.

Reviewer #3 (Recommendations for the authors):Figure 1:To which extent Thy1.1 expression correlates with IL-10 protein staining in the SG of MCMV-infected mice? This should be provided in the supplement.

We now show these data in Figure 1—figure supplement 1B.

Are all of these, or the majority of these cells, antigen-specific, i.e. what would be the correlation with for example available M25 tetramer staining or peptide pool restimulation?

This is an interesting point. Frustratingly, we do not believe that using M25 tetramers would help answer this question definitely in our non-TCR transgenic mouse system, given the breadth of the MCMV-specific CD4^+^ T cell response (*e.g.*, Arens *et al.*, J Immunol, 2008; Walton *et al.*, J Immunol, 2008), which extends to CD4^+^ T cells in the salivary glands that produce IL-10 (Clement *et al.*, PLoS Pathog, 2016). Moreover, it is clear from in vitro assays using PMA and ionomycin to stimulate CD4^+^ T cells isolated from the salivary glands that IL-10 is inefficiently detected via flow cytometry compared with Thy1.1 (Figure 1—figure supplement 1B), which makes even pooled peptide restimulation experiments difficult to perform with any degree of sensitivity. However, approximately 25% of Thy1.1^+^ CD4^+^ T cells coexpressed IFNγ in response to polyclonal stimulation (Figure 1—figure supplement 1E), suggesting they were derived from virus-specific Th1 cells, and even this figure is likely an underestimate of virus-specific CD4^+^ T cell frequencies based on data from CD4 transgenic mice (Jeitziner *et al.*, Eur J Immunol, 2013). It therefore seems likely that most Thy1.1^+^ CD4^+^ T cells are indeed specific for MCMV (now discussed on page 6).

It remains unclear to which extent the phenotype of in vivo generated IL-10+CD4^+^T cells overlaps with conventional in vivo generated Th1 cells (maybe even ones of acutely MCMV-infected mice), cytotoxic Th1 cells (for example lung cytotoxic Th1 cells) and Tr1-like cells (either generated in vitro with IL-27 and TGFb or isolated from an in vivo setting).

We have shown previously that IL-10-producing CD4^+^ T cells are not IL-27-inducible and are phenotypically distinct from Tr1 cells (Clement *et al.*, PLoS Pathogens, 2016). These cells also lack granzyme B and do not express genes that classically demarcate cytotoxic CD4^+^ T cells, such as Crtam and Klrc1, which encode NKG2A and NKG2D, respectively. Moreover, approximately 25% of Thy1.1^+^ CD4^+^ T cells expressed IFNγ in response to polyclonal stimulation (Figure 1—figure supplement 1E), and associated genes (*e.g.*, IL-18R) were downregulated among Thy1.1^+^ CD4^+^ T cells (*Figure 1D*). We also now show that IFNγ^+^ Thy1.1^+^ CD4^+^ T cells upregulate inhibitory receptors compared with IFNγ^+^ Thy1.1^−^ CD4^+^ T cells (Figure 1—figure supplement 1E). These data are consistent with the conclusion that IL-10-producing CD4^+^ T cells derive from Th1 cells generated in vivo but exhibit increased expression of secreted and membrane-bound inhibitory molecules and reduced production of IFNγ in response to cognate antigen stimulation. These observations are discussed on pages 4, 6, and 7.

Which gene families constitute 1646 DEG in Figure 1E.

Shared pathways are now available at https://doi.org/10.5281/zenodo.7447477. See also page 6.

The authors could also co-stain IFNγ and GzmB with Thy1.1 after polyclonal or even antigen-stimulation in vitro.

We failed to detect granzyme B expression via flow cytometry (Author response image 1), suggesting a disconnect between granzyme B message (RNAseq) and translation. This is now mentioned on page 6.

**Author response image 1. sa2fig1:** Granzyme B protein expression by salivary gland CD4^4^ is not detectable by flow cytometry. Granzyme B and Thy1.1 expression by salivary gland CD^4^ T cells was assessed by flow cytometry 14 days after MCMV infection of 10-Bit mice.

Some genes associated with the induction of IFN-g, including Il18r1 and il18rap, were downregulated in Thy1.1+ CD4^+^ T cells, however IFNγ does not seem to be differentially regulated, or? While Thy1.1+ cells express a module of inhibitory genes, their expression of Tcf1/Tcf7 seems to be downregulated?

Our data represent a snapshot of gene expression. It is therefore likely that some disconnect exists between certain patterns of gene and protein expression.

Is any difference in the expression of Tox observed?

*Tox1* is actually downregulated in Thy1.1^+^ cells (*Figure 1D*). We now highlight this finding on page 7.

Figure 2:The authors show that the Thy1.1+ pool shows a narrower and more convergent repertoire, which could be an indication of a clonally expanded population. VDJtools, which the authors use, offers a more comprehensive diversity estimation with PlotQuantileStats. This would enable the authors to visualize the repertoire clonality of Thy1.1+ vs Thy1.1- pools.

We have now included an analysis using PlotQuantileStats (Figure 2—figure supplement 2A).

For better understanding, is the clonality analysis performed on a sample derived of several pooled mice?

Clonality and other metrics were assessed using a minimum of four mice per group pooled across three independent experiments (m1, m2, and m3) as clarified in the legend for Figure 2.

Have the authors had the opportunity to assess the degree of clone sharing between e.g., splenic and salivary gland cells?

We have not performed this experiment. The kinetics of IL-10-producing CD4^+^ T cell responses at these sites are very different, and by day 14 p.i., there are very few IL-10-producing CD4^+^ T cells in the spleen (Clement *et al.*, PLoS Pathog, 2016). A direct comparison of these compartments at a single time point would therefore be difficult to undertake and possibly even more difficult to interpret meaningfully.

Figure 3:The authors demonstrate higher transcript levels and open chromatin structures of Arg1 in Thy1.1+ cells. They should provide a co-staining of Arg-1 and Thy1.1 on a protein level.

The intracellular detection of Arg1 via flow cytometry proved to be suboptimal in our analyses of C57BL/6 mice (discussed on page 13), but we have now included new data suggesting that Arg1 expression is indeed enriched among Thy1.1^+^ CD4^+^ T cells in the salivary glands (Figure 3C). We also show that only conditions that induce IL-10-producing CD4^+^ T cells induce the secretion of substantial amounts of Arg1 (Figure 3E). These data are consistent with the conclusion that the upregulation of Arg1 is a hallmark of inhibitory CD4^+^ T cells, which are also likely the predominant source of secreted Arg1 within the CD4^+^ T cell compartment. We discuss these data on pages 9, 10, and 15.

Although CD4 T cells are critical in controlling MCMV in the SG, Cd4cre mice target also CD8 and NKT cells can the authors confirm there is no Arginase-1 expression in these cells as well that could contribute to MCMV control in SG? This could potentially be solved by adoptive transfer of WT vs. Arg1-/- CD4 T cells. Otherwise, conclusion of this figure should be downplayed as it is not clear that Arg1 expression by CD4^+^ t cells selectively inhibit accumulation of virus-specific CD4^+^ and CD8^+^ T cells.

We now show that CD4 depletion but not CD8 depletion impacts Arg1 protein concentrations in salivary gland homogenates during chronic infection with MCMV (Figure 3—figure supplement 3B).

Is there any impact on the IL-10 expression by CD4^+^ T cells observed in the SG of CD4Cre Arg1flox mice infected with MCMV?

We now show that T cell-specific knockdown of Arg1 does not significantly impact virus-specific IL-10-producing CD4^+^ T cell responses, unlike virus-specific IFNγ^+^ CD4^+^ T cell responses (Figure 4D). These data suggest that Arg1 does not drive the development of these cells, as discussed on page 10.

NK cells are known to contribute to the viral control in SG and TRAIL+ NK cells capable of killing activated CD4^+^ T cells accumulate in the SG during CMV infection (Schuster et al. 2011). Do the authors observe any differences in the expression of DR5 in relation to Thy1.1?

We detected no differential expression of Trailr in Thy1.1^+^ CD4^+^ T cells versus Thy1.1^−^ CD4^+^ T cells, implying that the transient nature of these cells is likely not due to killing via TRAIL (https://doi.org/10.5281/zenodo.7243956). See also page 7.

Are there any functional and phenotypical advantages in the CD4^+^ T cell compartment of uninfected mice?

We now show that T cell-specific depletion of Arg1 has no obvious impact on the function or phenotype of CD4^+^ T cells in naive mice (Figure 3—figure supplement 3C, D).

The proliferation effect observed in CD4CreArg1flox mice – can it be reversed by Arginase-1 supplementation?

We do not believe that injecting Arg1 protein into mice will likely have a biologically meaningful impact on T cell responses in the salivary glands during chronic infection with MCMV. However, we have now inserted reference to human data demonstrating that Arg1 suppresses T cell proliferation (page 13).

Figure 4:In figure 3A, are the CD4 T cells pre-gated on CD44? One would assume that the majority of CD4^+^ T cells in the SG during CMV infection would be T-bet+, i.e., of Th1 phenotype. Can this be attributed to the technical issue of T-bet staining? Also does the FMO/Ab staining come from the same mouse as the % of Thy1.1 cells seem to increase from upper (11.9%) to lower (19.43%) plot?

We thank the reviewer for highlighting this issue. The representative plots were from different mice. The reviewer is probably correct that all activated CD4^+^ T cells will be T-bet^+^ in this model system. However, ex vivo T-bet staining is never particularly bright in our experience, particularly among T cells isolated from the salivary glands, and accordingly, we now show these data as histogram overlays with FMOs (Figure 5A).

Figure 5:In this figure the authors demonstrate that IFNγ-expressing cells are increased with T-bet deficiency and as a result the virus is cleared better. What is the correlation between IFN-g and Arg-1 expression?

The transient impact of T-bet depletion on viral replication likely represents a combination of factors, and accordingly, we do not necessarily expect to see a clear association between Arg1 and the expression of IFNγ. We have nonetheless shown that T-bet deficiency impinges on the development of virus-specific IL-10-producing CD4^+^ T cells (Figure 6A).

Are the virus specific IFNγ-producing cells in T-bet-deficient mice also of CD44loPD-1loCD62Lhi phenotype?

Our attempts to address this question using peptide-specific CD4^+^ T cells isolated from the salivary glands were inconclusive due to low overall cell numbers and low frequencies of virus-specific CD4^+^ T cells, as highlighted elsewhere. We therefore reverted to polyclonal stimulation, which unfortunately led to the downregulation of CD62L (see Author response image 2). We would be happy to attempt further peptide stimulations with much larger group numbers/pooled samples if the reviewer feels these data are essential to support the conclusions of our study. However, we would prefer not to conduct these experiments if the reviewer concurs, because the required number of mice would be considerable and therefore ethically questionable.

The title of the figure says that IL-10-producing CD4^+^ T cells develop in a T-bet-dependent manner, however the authors do not probe for IL-10 expression here.

We now include an analysis of IL-10-producing virus-specific CD4^+^ T cells, demonstrating reduced accumulation of these cells following the depletion of T-bet (Figure 6A).

What formal proof demonstrates that repetitive antigen exposure results in Arginase-1 upregulation? This could potentially be mimicked in vitro (Saraiva et al., 2009).

We thank the reviewer for this insightful suggestion. To address this issue, we stimulated OVA-specific transgenic CD4^+^ T cells with high-dose or low-dose antigen ± IL-12, which has been shown previously to induce the expression of IL-10 among Th1 cells (Saraiva *et al.*, Immunity, 2009). Only the combination of high-dose antigen ± IL-12 induced Arg1^+^ IL-10-producing cells (Figure 3D) and the secretion of substantial amounts of Arg1 (Figure 3E). These data suggest that high-dose antigen exposure acts similarly during chronic infection with MCMV. See also pages 9, 10, 15, and 23.

In Figure 5H, significance in viral clearance seems to be driven by a number of plaques observed in a sample of one mouse. Even like that, the difference does not reach even half a log a difference. I suggest increasing n to strengthen this result.

We have increased the numbers as requested by the reviewer, focusing the data on day 14 p.i. (now Figure 6I). The plaque assay data from the later time point are now shown in Figure 5—figure supplement 5D.

In contrast to Cd4Cre Tbet-flox mice which seem to compensate for the lack of T-bet by day 28 there is no protective effect, viral control is still observed in Cd4Cre Arg1flox (this paper) and CD4Cre IL10-flox (author´s previous publication Clement et al., 2016). How do the authors comment on this?

We comment on this issue on page 12.

[Editors' note: further revisions were suggested prior to acceptance, as described below.]

Reviewer #2 (Recommendations for the authors):In the revision, the authors have performed all requested experiments.I do however not think that the interpretation of their data is acceptable in its current form, because they ignore a large body of published evidence on IL-10 producing T-cells that is highly relevant to their study. There is already a lot of confusion in the field, and ignoring seminal work performed by others does not contribute to reaching a general consensus on the molecular identity of these cells.The authors show in supplementary Figure 5 that IL-10 producing regulatory T-cells express high levels of Eomesodermin, a rather surprising finding for CD4^+^T-cells, but comment on this data only briefly on p11: "No such difference in expression frequency were observed for the related transcription factor Eomesodermin (Eomes)". This is really a misleading statement since the authors do show that the MFI of Eomes is higher in IL-10 producing T-cells. However, no statistical analysis is provided, and a control with CD4^+^T-cells in the absence of CMV infection (presumably Eomes-negative) is also missing. In any case, the large majority of IL-10+ T-cells express high levels of Eomesodermin, while the staining of T-bet suggests that T-bet expression may be actually lower (of course this may be due to technical issues). Thus, the choice to focus on T-bet rather than on Eomes is in my opinion an arbitrary one. I understand that it would take too much work to perform all relevant experiments both with T-bet and Eomes-deficient T-cells in a revision, but ignoring all the published evidence on Eomesodermin-expressing, IL-10 producing CD4^+^T-cells is simply unacceptable.

We have now expanded our introduction of this data, quoting on page 11 that we do not see a significant difference in Eomes protein between Thy1.1^+^ and Thy1.1^-^ CD4^+^ T cells (data shown in Author response image 3 and quoted on page 11). We discuss the potential relevance of EOMES in the development of SG CD4^+^IL10+ T cells on page 16.

**Author response image 3. sa2fig3:** Eomes expression by Thy1. 1^+^ and Thy1.1- CD4^+^ T cells 14 days post-MCMV infection.

The authors state moreover in the Introduction that IL-10 producing T-cells in their system "are phenotypically different from type 1 regulatory T-cells (Clement 2016)" In their cited work I could not find a detailed phenotypic characterization, so it is unclear to me how the authors came to this conclusion.

We have revised this statement for improved clarity (page 4).

Moreover, in the revised article, they document in contrast that the IL-10 producing "inhibitory" T-cells they study do express all markers that are well-known to be associated with Tr1-cells, namely LAG3, PD1 and other checkpoint receptors, CCR5, GzmK etc etc.(Brockmann, Soukou et al. 2018, Roessner, Llao Cid et al. 2021, Thelen, Schipperges et al. 2023) Thus, the relationship of the IL-10 producing "inhibitory" T-cells to Tr1-cells, in particular those that express Eomes, needs to be discussed in detail and the relevant articles have to be cited. Notably, also Eomes-expressing Tr1-cells are T-bet dependent (Zhang, Lee et al. 2017) and clonally expanded (Bonnal, Rossetti et al. 2021).

We thank the reviewer and have inserted a paragraph on pages 15-16 to address these important studies.